# Sequence-to-sequence models with attention mechanistically map to the architecture of human memory search
Nikolaus Salvatore [1] & Qiong Zhang [1,2,3] ✉

Past work has long recognized the important role of context in guiding how humans search their memory. While context-based memory models can explain many memory phenomena, it remains unclear why humans develop such architectures over possible alternatives in the first place. In this work, we demonstrate that foundational architectures in neural machine translation – specifically, recurrent neural network (RNN)-based sequence-to-sequence models with attention – exhibit mechanisms that directly correspond to those specified in the Context Maintenance and Retrieval (CMR) model of human memory. Since neural machine translation models have evolved to optimize task performance, their convergence with human memory models provides a deeper understanding of the functional role of context in human memory, as well as presenting alternative ways to model human memory. Leveraging this convergence, we implement a neural machine translation model as a cognitive model of human memory search that is both interpretable and capable of capturing complex dynamics of learning. We show that our model accounts for both averaged and optimal human behavioral patterns as effectively as context-based memory models using a publicly available free recall experiment dataset involving 171 participants. Further, we demonstrate additional strengths of the proposed model by evaluating how memory search performance emerges from the interaction of different model components.

As humans and machine learning systems often face similar computational challenges[1], there has been synergy between machine learning and cognitive science research, leveraging machine learning advancements to better model human cognition[2,3] and cognitive models to inform better design of intelligent systems[4,5]. Identifying parallels between models in machine learning and models of human cognition is critical to the interplay between these two fields. In this work, we make connections between two prominent classes of models previously developed from these two separate communities. In machine learning, one cornerstone of neural machine translation is the development of sequence-to-sequence (seq2seq) models to handle variable-length input sequences in natural language processing tasks[6,7]. Their efficiency is further improved by the attention mechanism[8,9], which laid the foundation for the now ubiquitous Transformer models[10–13]. In cognitive science, researchers construct and test mathematical models of human memory to understand how information is encoded and retrieved. Decades of empirical work have recognized the important role of context in encoding and later guiding the retrieval of information[14–18]. This notion of context has been formally captured in frameworks such as the context

maintenance and retrieval (CMR) model, which can explain a wide range of memory behavioral patterns during free recall[17,19–21], serial recall[22,23], free association[24], collaborative recall[25], as well as a broader range of behavior in memory consolidations[26], rewards[27], and decision making[28].

Despite the different forces driving these model developments (task performance in machine learning versus alignment with human behavioral data in cognitive science), we demonstrate in this work that neural network models in neural machine translation (specifically the RNN-based seq2seq models with attention) and context-based models of human memory (specifically the CMR model) exhibit strikingly similar architectural components. To uncover this relationship, we review the historical developments of models in the two fields and highlight how two major advancements in each field are analogous to each other. We also provide a detailed mathematical mapping showing that the sequence-to-sequence (seq2seq) architecture in neural machine translation corresponds to how information is encoded and later accessed in human memory, and that the attention mechanism corresponds to how humans reactivate prior mental contexts. Identifying the convergence between the neural machine

¹Rutgers University-New Brunswick, Department of Computer Science, Piscataway, NJ, USA. ²Rutgers University-New Brunswick, Department of Psychology, Piscataway, NJ, USA. ³Rutgers Center for Cognitive Science, Piscataway, NJ, USA. ✉e-mail: qiong.z@rutgers.edu

translation model and the context-based human memory model has two important implications.

First, the convergence between the neural machine translation model and the CMR model provides a deeper understanding of the functional role of context in human memory. Despite the success of the CMR model in explaining various behaviors in human memory[17,19,20], we do not yet know if there is a functional role in utilizing context to encode and retrieve information. Does human memory rely on the use of context to maximize the chance of retrieving the correct target information? A rational approach[29,30] to address this question is to iterate through all possible architectures in describing representations and processes of memory search and identify which among them achieves optimal task performance. Though exhaustively analyzing this vast architectural space in a single study is impractical, we argue that the trajectory of model development in neural machine translation follows exactly such an analysis, as the field explores the space of architectures in search of those that optimize task performance. The convergence between the development of the CMR model (driven by alignment with human behavioral data) and the development of neural machine translation models (driven by task performance) provides evidence that the architectural assumptions specified in CMR serve an adaptive purpose. In other words, it may be rational for human cognition to organize and retrieve information in this specific way using context, and alternative architectural constraints could be less effective in serving the goal of the memory system.

Second, the convergence between the neural machine translation model and the CMR model opens up alternative ways of modeling context in human memory. Traditional cognitive models like CMR use mathematical equations and a small set of parameters to describe how people search their memories using context. These models are effective in describing qualitative memory behaviors through parameter fitting, but are limited in their ability to explain how different behavioral patterns emerge from the process of learning and contribute to memory performance. By contrast, neural network models offer such model flexibility, but often sacrifice interpretability due to their numerous parameters and complex model architectures. Identifying the parallels between architectural components in a neural network model of machine translation and a mathematical model of context memory lays the foundation for building a model that is both flexible (being a neural network model) and interpretable (has connections to known cognitive components). In this work, we implement such a neural network model of machine translation as a cognitive model of human memory search. We first demonstrate that a basic seq2seq model with attention[8] can capture and predict human recall patterns as effectively as CMR in a free recall task. Next, we train our model in a reinforcement learning framework to examine the emergent behavioral patterns during the learning process and after task performance is optimized. Our results indicate that the fully trained seq2seq model with attention aligns with the behavioral patterns of optimal free recall, as produced by the rational analysis of the Context Maintenance and Retrieval model (rational-CMR)[31]. Furthermore, model evaluations conducted intermittently throughout the model's training exhibit similar recall characteristics as human participants in terms of recency and temporal contiguity effects[32–34].

While the above analyses aim to establish seq2seq models as an alternative model of human memory search comparable to CMR, we conduct modeling analyses to demonstrate the additional strengths of the seq2seq model with attention. We examine the effect of working memory capacity on recall strategies, as implemented as a change in the hidden state dimension size of the seq2seq model. We show that reduced working memory capacity requires a compensatory mechanism and a stronger reliance on using context retrieved from episodic memory in order to support memory performance. We also examine the role of episodic memory through an ablation study of the attention mechanism, with the goal of better understanding memory deficiencies in hippocampal amnesia. Prior research has linked medial temporal lobe (MTL) damage to an inability to reinstate previously experienced contexts[35–38]. As we identify a parallel between the attention mechanism in our seq2seq model and how humans reactivate prior experienced contexts, ablation of the attention mechanism should capture recall characteristics similar to those of patients with hippocampal amnesia compared with healthy controls.

In the following sections of the paper, we first identify the convergence of the seq2seq model with attention and the CMR model by reviewing their historical developments and by providing a detailed mathematical mapping between their architectural components. Next, we implement a basic seq2seq model with attention as a cognitive model of human memory search as studied in a free recall task. We detail our evaluation results across a range of model configurations, demonstrating the model's ability to capture and predict human free recall behavior, as well as providing insights into how different model components interact and contribute to memory performance.

## Methods
### Identifying parallels in historical developments
**The sequence-to-sequence model architecture and early formulations of human context models**. We will start by reviewing the historical developments of models in the two fields and highlight how two major advancements in each field are analogous to each other, despite their developments being driven by different forces (task performance in machine translation and alignment with human data in cognitive science). Early approaches in machine translation relied on more stringent and limited rule-based methods to perform automated translation. The oldest of approaches, rule-based methods, involve applying hand-crafted linguistic rules that perform well in limited scenarios, but struggle to extrapolate to the complexity and variability of natural language[39]. Statistical machine translation greatly improved the performance of rule-based methods, leveraging word-based and later phrase-based statistical methods drawing from large bodies of bilingual text[40,41]. These earlier techniques were largely supplanted by neural network-based methods of machine translation (termed neural machine translation) that displayed unparalleled flexibility and spawned an entire subfield of research. Despite the success of deep neural networks in a range of difficult problems such as speech recognition[42] and visual object recognition[43], early development of neural network models, including recurrent neural networks suitable for sequence modeling (RNNs[44]), can only be applied to problems whose inputs and targets can be encoded in vectors of fixed dimensionality. This poses a serious limitation in many tasks, such as machine translation, that are expressed with sequences whose lengths are not known in advance. The development of the sequence-to-sequence (seq2seq) model was a milestone in neural machine translation, enabling more effective handling of variable-length input and output sequences. In these models, an encoder RNN maps a variable-length source sequence (indicated in Fig. 1A) to a fixed-length context vector, $h_L$, which is then subsequently used by a decoder RNN to generate a variable-length target sequence[6,7].

The encoding and decoding processes found in seq2seq models closely resemble how information is encoded and recalled in context-based models in human memory. Human memory search, when studied in a free recall task, is also a sequence modeling task, where the input sequence is a list of items presented and the output sequence is the list of items recalled (illustrated in Fig. 1B). Many behavioral findings in free recall literature can be captured by a model where a contextual representation slowly drifts over time and is associated with to-be-remembered experiences[17,19,20]. The context at the end of the encoding period is carried over to the recall period if the delay or the distractor task between study and recall is minimal. During recall, the contextual representation serves as a cue to drive a sequence of recalls. Different than the development of machine translation models, which has been driven by the need to perform well on the translation task, the development of context-based models in human memory has been driven by the need to account for behavioral patterns in human data and can be traced back to Bower's (1972) temporal context model[45,46]. Since memory retrieval success is a function of context overlap between study and recall, Bower's temporal context model can account for how we forget by assuming a context vector that drifts randomly and becomes more dissimilar over

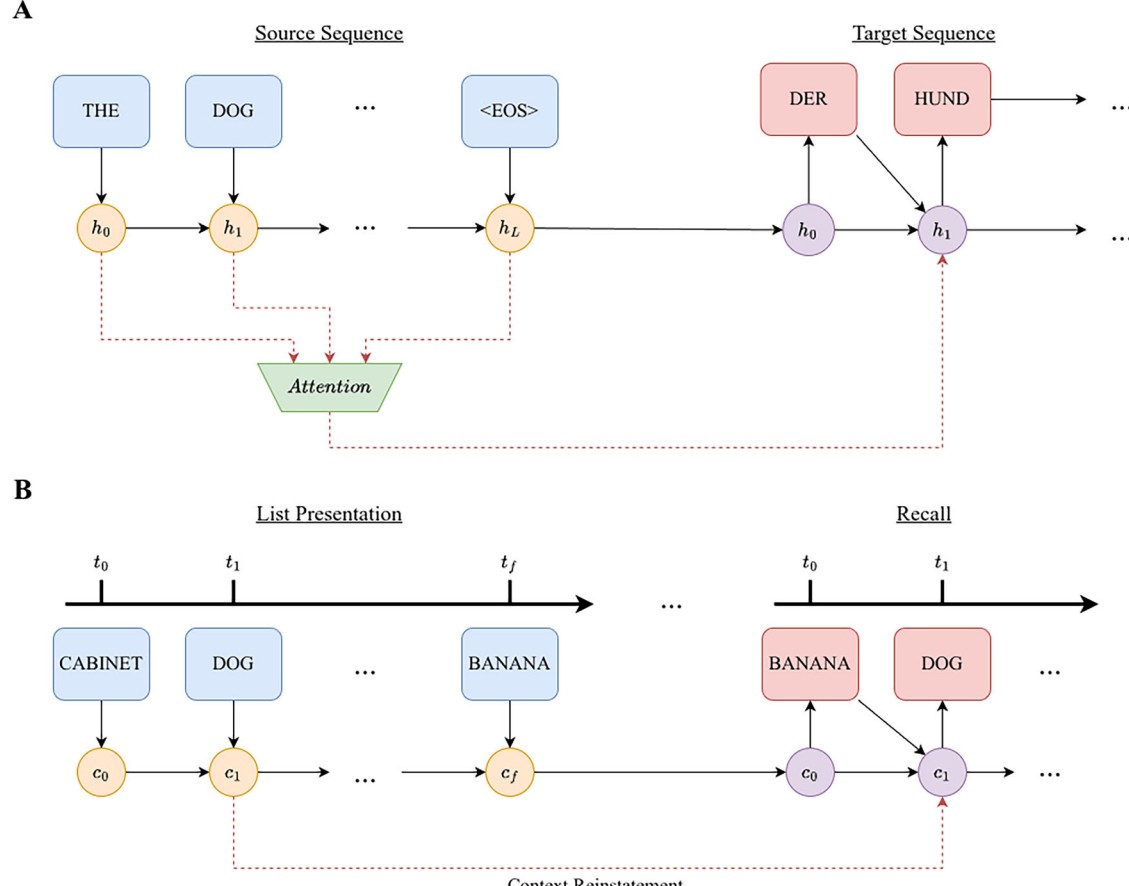

**Fig. 1 | Illustrating the parallels between neural machine translation and human memory search models. A** A seq2seq model with attention begins machine translation tasks by encoding each word of the original sequence into its hidden state, updated iteratively at each timestep, using its RNN encoder. In the decoding stage, the decoder RNN receives the final hidden state from the encoding stage and generates a word in the target language. The attention mechanism gives the decoder RNN direct access to each encoder's hidden state at each decoding step. After each word is translated, it is fed back into the decoder, along with its updated hidden state, to iteratively generate a target sequence. **B** A context-based model of the human free recall task begins with the model encoding each word of the presented list into its latent context. During recall, participants try to recall as many of the words from the list as possible in any order. This recall process is driven by the similarity between the encoding and recall contexts, starting with the final context from the list presentation. Through the context reinstatement mechanism, the original encoding context of a just-remembered item is reactivated, similar to how the attention mechanism reactivates previous encoding hidden states.

time. The model can also account for how one recognizes items studied on different lists, as items within a given list will have more overlap in their contexts. Bower's formulation of context marks an important advancement in the field and provides a foundation for more recent computational models of human memory search[17,19,20].

**The attention mechanism and the context reinstatement mechanism.** The second major advancement in neural machine translation is the attention mechanism. Seq2seq models enabled the processing of variable-length input sequences into fixed-length vectors[6,7]. However, this approach initially struggled with long sentences, as the fixed-length context vector limits the amount of contextual information that could be embedded into the encoder's output. To address this limitation, the attention mechanism[8,9] allows the model to focus on different parts of the input sequence when generating each word in the target sequence. The mechanism helps capture dependencies between distant words more effectively by giving the decoder RNN direct access to previous encoder states during the decoding process. Each hidden state of the encoding stage is assigned an attention weight by the mechanism and used to inform each step of the decoding stage (as indicated in Fig. 1A). These developments laid the foundation for more advanced architectures, including the now ubiquitous Transformer model[10–13].

We draw a parallel between the attention mechanism and the context reinstatement mechanism in context-based models of human memory.

Early formulations of context saw context as a cue for items but did not explicitly consider that items can alter context[45,46]. More recent computational models of human memory search, such as the CMR[19,20], a successor of the Temporal Context Model[17], introduced the idea that remembering an item reinstates its original context at the time of encoding. As illustrated in Fig. 1B, remembering the word "dog" calls back its original encoding context, $c_1$. This important addition to context-based models is driven by the human recall pattern: items studied close to each other are likely to be recalled together (temporal contiguity effect[34]). The method by which CMR reactivates previous encoding contexts is analogous to how an attention mechanism can reactivate previous encoding hidden states. Once items tied to the present context are depleted, context reinstatement enables a "jump back in time" to earlier study contexts (akin to Tulving's concept of mental time travel[47]), providing additional retrieval cues to recall the remaining items.

**Deriving a detailed mathematical mapping**
In addition to highlighting the parallels in their historical developments, in this section, we provide a detailed mathematical mapping between the architectural components of seq2seq models with attention and the CMR model. As illustrated in Fig. 2, we align the encoding and decoding (or recall) processes across both frameworks, detailing each step. Specifically, Fig. 2A, C show the encoding phases for the seq2seq model and the CMR model, respectively, while Fig. 2B, D depict their corresponding decoding

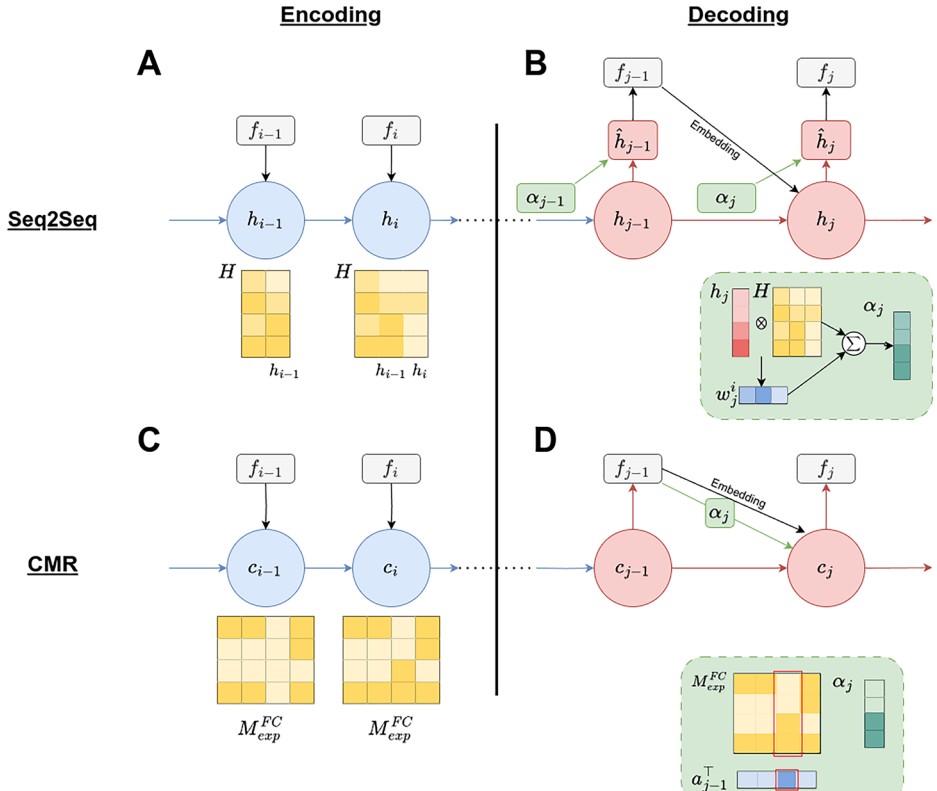

**Fig. 2 | Detailed mapping between components in the seq2seq model with attention and those in the CMR Model.** During the encoding phase, both the seq2seq model with attention (**A**) and the CMR model (**C**) maintain an internal state (hidden state $h_i$ or context vector $c_i$) that combines input features of a presented item ($f_i$) and the previous encoding state ($h_{i-1}$ or $c_{i-1}$) at each step $i$. In CMR, two fixed parameters control the mixing of the previous encoding state and the new item embedding, while the seq2seq model accomplishes the same task through more complex parameter matrices. During the decoding or recall phase, both the seq2seq model with attention (**B**) and the CMR model (**D**) use the current decoding state to drive the output of the next item. Importantly, the decoding state that drives the next output, $\hat{h}_j$ in the seq2seq model, contains three components equivalent to those of $c_j$ in CMR: i) the hidden state or context state from the previous decoding step ($h_{j-1}$ or $c_{j-1}$; red arrow), ii) an input embedding ($f_{j-1}$) of the most recently recalled item (black arrow), and iii) reactivated hidden states or contexts states ($\alpha_j$) from the encoding phase (green arrow). The attention mechanism (indicated in the green inset) of the seq2seq model computes an attention context vector $\alpha_j$, which is calculated as a sum of the encoding contexts weighted by the similarities between the current decoding state $h_j$ and previous encoding states stored in $H$. This attention mechanism is mathematically equivalent to the context reinstatement mechanism in CMR, also denoted here as $\alpha_j$, by applying the activation strength of the previous recall step $a_{j-1}$ to the previous encoding contexts stored in $M_{exp}^{FC}$. The seq2seq model with attention uses Gated Recurrent Units[6] and the Luong attention mechanism[8], and the CMR model follows implementations in the previous work[17,19,31]. Color coding highlights the functional correspondence between components in the two models.

and recall phases. We will first describe each model in detail, then derive the mathematical mapping between them. Our model descriptions closely follow (and are mathematically equivalent to) existing implementations of seq2seq models with attention[8] and CMR[17,19,31], though the exact notation has been slightly modified to facilitate easy visualization and alignment of the two models.

### Seq2seq model with attention

**Encoding phase.** The encoding phase of the seq2seq model, pictured in Fig. 2A, is analogous to the study phase in a memory experiment, where participants encode a list of items into an evolving mental context. Each input item presented at timestep $i$ is embedded into a dense vector $x_i \in \mathbb{R}^d$ using pre-trained embedding vectors (in our case, GloVe embeddings[48]). The encoder RNN processes the sequence one item at a time, updating its hidden state $h_i \in \mathbb{R}^d$ at each time step according to the equation below:

$$h_i = \phi\left(W_h x_i + U_h h_{i-1} + b_h\right) \quad (1)$$

In Eqn. (1), $W_h$ and $U_h$ are learned parameter matrices, $b_h$ is a learned bias vector, $\phi$ is a non-linear activation function such as tanh, sigmoid, etc., and $h_{-1}$ is the hidden state from the previous timestep. In this process, the weight matrix $U_h$ controls the degree to which the previous hidden state $h_{i-1}$ is

maintained in the current hidden state $h_i$, while the weight matrix $W_h$ controls the degree to which the embedding of the newly presented item, $x_i$, is incorporated into the current hidden state.

In addition to updating the hidden state $h_i$, the model also stores the concatenation of all hidden states, $H = [h_1, h_2, h_3, …, h_L]$ where $L$ is the number of steps/items during encoding. These states $H$ later inform the decoding process to access the encoder states through the attention mechanism.

**Decoding Phase.** The decoding phase, pictured in Fig. 2B, is initialized with the final hidden state from the encoding phase, i.e., $h_L$. During each decoding step $j$ (which we distinguish from encoding step $i$), the decoder RNN receives the embedding $x_j \in \mathbb{R}^d$ of the just-recalled item from the previous step $j - 1$, along with the previous hidden state $h_{j-1} \in \mathbb{R}^d$:

$$h_j = \phi\left(W_{h'} x_j + U_{h'} h_{j-1} + b_{h'}\right) \quad (2)$$

Eqn. (2) is identical to Eqn. (1) used in the encoding phase but employs different parameters $W_{h'}$, $U_{h'}$, and $b_{h'}$ learned for the decoding phase.

During decoding, the attention mechanism (shown in the green box in Fig. 2B) gives the decoder access to all encoder hidden states $H$ via attention weights. These attention weights are obtained by calculating a score function

between the current decoder hidden state $h_j$ and each hidden state $h_i$ from the encoding stage. We use the dot product as the score function as specified in Luong attention[8] to avoid introducing additional parameters. After calculating scores for each encoder hidden state, the softmax operation is applied to obtain the attention weights $w$ as shown below:

$$w_j^i = \frac{\exp\left(h_j^\top h_i\right)}{\sum_{i=1}^{L} \exp\left(h_j^\top h_i\right)} \quad (3)$$

where $h_j^\top h_i$ is the dot product, i.e., the similarity between the current decoder hidden state $j$ and an encoder hidden state $h_i$. The attention weight $w_j^i$ refers to how much attention the model should pay to each encoder hidden state $h_i$ relative to all other encoding states at the decoding step $j$, which represents the importance of the encoder hidden state $h_i$ to producing the output at the decoding step $j$.

Once the attention weights have been computed, the overall attention context vector ($\alpha_j$) is obtained via a weighted sum of encoder states:

$$\alpha_j = \sum_{i=1}^{L} w_j^i h_i \in \mathbb{R}^d \quad (4)$$

Before an output is generated, a final hidden state vector $\hat{h}_j$ is formed by combining the decoder hidden state and the attention context through a dense, mixing layer:

$$\hat{h}_j = \tanh\left(W_c\left[h_j; \alpha_j\right]\right) \quad (5)$$

Here, $W_c \in \mathbb{R}^{dx2d}$ are the learned parameters of the mixing layer, while $[h_j; \alpha_j]$ is the concatenation of the current decoder hidden state $h_j$ and the attention context vector $\alpha_j$.

An output, $f_j$, is generated at the decoding step $j$ based on an output/retrieval rule $\psi$ in conjunction with the final hidden state $\hat{h}_j$:

$$f_j = \psi(\hat{h}_j) \in \mathbb{R}^N \quad (6)$$

where $f_j \in \mathbb{R}^N$ is a one-hot column vector that is all zeros except at the position representing the item's identity, and $N$ is the total number of possible items in the experiment.

## CMR model

**Encoding phase**. During the encoding phase in the free recall task (pictured in Fig. 2C), participants study a list of $L$ items one after another (drawn from a total number of $N$ possible items in the experimental word pool). The CMR model proposes that their context slowly drifts towards the memory representations of recently encountered experiences. The state of the context at time step $c_i \in \mathbb{R}^N$ is given by:

$$c_i = \rho\, c_{i-1} + \beta\, x_i \quad (7)$$

where $x_i \in \mathbb{R}^N$ is the retrieved context (or input embeddings) of the just-encoded item, $\beta \in [0, 1]$ is a parameter determining the rate at which context drifts toward the new context, and $\rho$ is a scalar ensuring $||c_i|| = 1$. The retrieved context $x_i$ is further expressed as:

$$x_i = M_{pre}^{FC} f_i \quad (8)$$

where $M_{pre}^{FC} \in \mathbb{R}^{N \times N}$ represents item-to-context associations that existed prior to the experiment (initialized as an identity matrix, under the simplifying assumption that an item is only associated with its own context[19]), and $f_i \in \mathbb{R}^N$ is a one-hot column vector that is all zeros except at the position that represents an item's identity. Therefore, $M_{pre}^{FC} f_i$ is the context previously associated with the presented item at encoding step $i$, which is simply $f_i$.

In addition to updating the context $c_i$, CMR forms associations between items and the evolving context throughout the encoding phase to capture new learning in the experiment – through experimental item-to-context and context-to-item associations held in $M_{exp}^{FC} \in \mathbb{R}^{N \times N}$ and $M_{exp}^{CF} \in \mathbb{R}^{N \times N}$. These matrices will be useful later in the recall phase to reactivate the corresponding encoding context of a given item ($M_{exp}^{FC}$) or to retrieve an item corresponding to a given context ($M_{exp}^{CF}$). They are initialized to zero at the start of the experiment and are updated via the Hebbian outer-product learning rule. Specifically, when an item is encoded at timestep $i$, an association is formed between the previous context state $c_{i-1}$ and the presented item $f_i$:

$$\Delta M_{exp}^{FC} = c_{i-1} f_i^\top \quad (9)$$

Similarly, the association from context to item, $M_{exp}^{CF}$, is updated according to:

$$\Delta M_{exp}^{CF} = f_i c_{i-1}^\top \quad (10)$$

Following equations (9) and (10), after all $L$ items in a list have been studied, the final experimental association matrices before the start of the recall phase can be written as:

$$M_{exp}^{FC} = \sum_{i=1}^{L} c_{i-1} f_i^\top \quad (11)$$

$$M_{exp}^{CF} = \sum_{i=1}^{L} f_i c_{i-1}^\top \quad (12)$$

Together, Eqns. (7), (11) and (12) captures how CMR embeds each item into a gradually drifting context space, binding together items encountered close together in time and allowing the model to capture temporal contiguity effects observed in free recall[34]. To illustrate $M_{exp}^{FC}$ and $M_{exp}^{CF}$ defined in Eqns. (11) and (12) with an example, consider a short list of $L = 3$ items during encoding $f_1^\top = [0, 1, 0, \ldots]$, $f_2^\top = [1, 0, 0, \ldots]$ and $f_3^\top = [0, 0, 1, \ldots]$, which are associated through Hebbian learning with contexts at the preceding step $c_0$, $c_1$ and $c_2$ respectively ($c_0$ denotes the context vector prior to any encoding). After all $L$ items in the list are encoded, the resulting association matrices take the following form $M_{exp}^{FC} = [c_1, c_0, c_2, \ldots]$, where each context vector is stored in the column corresponding to the encoded item's identity. Similarly, the context-to-item association matrix is given by $M_{exp}^{CF} = [c_1^\top; c_0^\top; c_2^\top; \ldots]$, where each context vector is stored in the row corresponding to the encoded item's identity.

**Recall phase**. The recall phase of CMR, pictured in Fig. 2D, begins with the final context vector carried over from the encoding phase. On each recall step $j$, the context vector $c_j$ is updated to reflect both the influence from the just-recalled item $f_{j-1}$ (controlled by $\beta'$) and the context from the previous timestep $c_{j-1}$ (controlled by $\rho'$), similarly to how context drifts during the encoding phase in Eqn. (7):

$$c_j = \rho' c_{j-1} + \beta'\left[(1 - \gamma_{FC})x_j + \gamma_{FC}\alpha_j\right] \quad (13)$$

However, different than the encoding phase, the reactivated context from the just-recalled item $f_{j-1}$ comprises not only its pre-experimental context $x_j = M_{pre}^{FC} f_{j-1}$ (i.e., input embeddings, black arrow in Fig. 2D), similarly to Eqn. (8), but also the experimental context $\alpha_j$ (green arrow in Fig. 2D) which is given by:

$$\alpha_j = M_{exp}^{FC} f_{j-1} \quad (14)$$

The experimental context $\alpha_j$ retrieves the context associated with the item $f_{j-1}$ during encoding through applying the experimental item-to-context associations $M_{exp}^{FC}$. The extent of retrieving an item's

pre-experimental context $x_j$ versus experimental context $\alpha_j$, as shown in Eqn. (13), is determined by a parameter, $\gamma_{FC} \in [0, 1]$.

To determine which item $f_j$ is to be retrieved at timestep $j$, CMR examines how much the current context $c_j$ matches with all items' experimental contexts, as stored in rows of $M_{exp}^{CF} = \sum_{i=1}^{L} f_i c_{i-1}^{\top}$ (Eqn. (12)):

$$a_j = M_{exp}^{CF} c_j = \left( \sum_{i=1}^{L} f_i c_{i-1}^{\top} \right) c_j = \sum_{i=1}^{L} f_i (c_{i-1}^{\top} c_j) \qquad (15)$$

Here, $a_j \in \mathbb{R}^N$ is the activation strength, where each element reflects the similarity between the context at which item $i$ was encoded, $c_{i-1}$, and the present context $c_j$ during recall. Intuitively, items whose original encoding context more closely matches the current context are more likely to be recalled. Items that did not appear during the encoding phase have zero activation strengths.

To translate the activation strength $a_j$ into recall probabilities, the model applies a softmax function over the subset of elements in $a_j$ corresponding to items that appeared during the encoding phase. The probability of recalling the item from the encoding step $i$, $f_i$, at recall step $j$ is given by:

$$P(f_j = f_i) = \frac{\exp\left[ k\, c_{i-1}^{\top} c_j \right]}{\sum_{i'=1}^{L} \exp\left[ k\, c_{i'-1}^{\top} c_j \right]} \qquad (16)$$

Here, $k$ is a parameter that determines the amount of noise present in the retrieval process, where a higher value of $k$ favors a more noiseless recall and a higher chance of retrieving the item with the strongest activation. Finally, the recalled item at timestep $j$, $f_j$, is sampled from a multinoulli distribution whose probabilities for possible outcomes are specified in Eqn. (16).

### Putting everything together: mapping the seq2seq model with attention to CMR

**Encoding phase**. Both the encoding and decoding/recall phases of the seq2seq model with attention and CMR exhibit deep functional parallels. The correspondence between the two models during the encoding phase can be easily seen from Eqn. (1) and Eqn. (7). During each step of the encoding phase, both models update their current hidden state $h_i$ or context state $c_i$ by taking in an input embedding, $x_i$, that incorporates pre-experimental information of the presented item $f_i$, which is then combined with the hidden state $h_{i-1}$ or context state $c_{i-1}$ from the previous encoding step. In CMR, two parameters, $\rho$ and $\beta$, control the mixing of the previous context and the new item embedding, while the seq2seq model accomplishes the same task through more complex parameter matrices $W_h$ and $U_h$ together with the bias vector $b_h$ and the activation function $\phi$.

**Decoding/recall phase**. Compared with the encoding phase, the parallels between the seq2seq model and CMR for the decoding phase are less straightforward and require additional derivations, which we will demonstrate in detail below. The key function during the decoding phase for both models is to generate an output or recall at each timestep $j$. While the seq2seq model with attention uses the hidden state $\hat{h}_j$ to determine the output item $f_j$ (as shown in Eqn. (6) and Fig. 2B), the CMR model utilizes its context state $c_j$ to determine the recalled item $f_j$ (as shown in Eqn. (16) and Fig. 2D). Our primary goal here is to demonstrate the correspondence in the decoding phase between these two models, specifically by showing that $\hat{h}_j$ in the seq2seq model contains components equivalent to those of $c_j$ in CMR.

Combining Eqn. (1) and Eqn. (5), $\hat{h}_j$ in the seq2seq model with attention can be written as:

$$\hat{h}_j = \tanh\left( W_c \left[ \phi\left( W_{h'} x_j + U_{h'} h_{j-1} + b_{h'} \right); \alpha_j \right] \right) \qquad (17)$$

According to Eqn. (13), $c_j$ in CMR is given by:

$$c_j = \rho' c_{j-1} + \beta' \left[ (1 - \gamma_{FC}) x_j + \gamma_{FC} \alpha_j \right] \qquad (18)$$

Examining Eqn. (17) and Eqn. (18), it is clear that both $\hat{h}_j$ and $c_j$ share two common components: the hidden state or context state from the previous decoding step ($h_{j-1}$ or $c_{j-1}$) and an input embedding ($x_j$) of the most recently recalled item. Crucially, $c_j$ and $\hat{h}_j$ also incorporate a third component, $\alpha_j$, which are reactivated hidden states or contexts from the encoding phase. For the remainder of the derivation, we will show that these reactivated hidden states $\alpha_j^{RNN}$ (from the seq2seq model's attention mechanism) are equivalent to the reactivated contexts $\alpha_j^{CMR}$ (from the CMR model's context reinstatement mechanism).

Combining Eqn. (11) and Eqn. (14), we can write the reactivated context in CMR as:

$$\alpha_j^{CMR} = M_{exp}^{FC} f_{j-1} = \left( \sum_{i=1}^{L} c_{i-1} f_i^{\top} \right) f_{j-1} \qquad (19)$$

Under the condition that the most recently recalled item at the decoding step $j-1$ is the item studied at the encoding step $i$, i.e., $f_{j-1} = f_i$, Eqn. (19) can be written as:

$$\alpha_j^{CMR} = \left( \sum_{i=1}^{L} c_{i-1} f_i^{\top} \right) f_i = c_{i-1} \quad \text{if } f_{j-1} = f_i \qquad (20)$$

Since item recall is probabilistic, the identity of the just-recalled item $f_{j-1}$ is drawn from the distribution defined by the previous context state $c_{j-1}$, which can be written as (following Eqn. (16)):

$$P(f_{j-1} = f_i) = \frac{\exp\left[ k\, c_{i-1}^{\top} c_{j-1} \right]}{\sum_{i'=1}^{L} \exp\left[ k\, c_{i'-1}^{\top} c_{j-1} \right]} \qquad (21)$$

With Eqn. (20) and Eqn. (21), we can now write the expected value of $\alpha_j^{CMR}$ as below,

$$\mathbb{E}\left[ \alpha_j^{CMR} \right] = \sum_{i=1}^{L} \frac{\exp\left( k c_{i-1}^{\top} c_{j-1} \right)}{\sum_{i'=1}^{L} \exp\left( k c_{i'-1}^{\top} c_{j-1} \right)} c_{i-1} \qquad (22)$$

The expected value of context reinstatement in CMR, $\mathbb{E}[\alpha_j^{CMR}]$, is directly analogous to the attention context vector in the seq2seq model. Combining Eqn. (3) and Eqn. (4), we can write $\alpha_j^{RNN}$ in the seq2seq model as:

$$\alpha_j^{RNN} = \sum_{i=1}^{L} w_j^i h_i = \sum_{i=1}^{L} \frac{\exp\left( h_j^{\top} h_i \right)}{\sum_{i'=1}^{L} \exp\left( h_j^{\top} h_{i'} \right)} h_i = \sum_{i=1}^{L} \frac{\exp\left( h_i^{\top} h_j \right)}{\sum_{i'=1}^{L} \exp\left( h_{i'}^{\top} h_j \right)} h_i \qquad (23)$$

Examining Eqn. (22) against Eqn. (23), we can establish that the seq2seq model's attention mechanism is equivalent to the CMR model's context reinstatement mechanism, i.e., $\mathbb{E}[\alpha_j^{CMR}] \approx \alpha_j^{RNN}$. Intuitively, both models use a current decoding state ($h_j$ or $c_{j-1}$) as a probe to reactivate a weighted average of states from the encoding phase based on their similarity. The retrieved encoding states allow the model to access relevant information in the past to guide the generation of the next item. While we only examined Luong attention in this derivation, alternative implementations of attention[9] are analogous to Luong attention but with a different score function to quantify similarity. This concludes our proof for the decoding phase, where we demonstrate that what drives the next recall, $\hat{h}_j$ in the seq2seq model and $c_j$ in CMR, contains equivalent components as specified in Eqn. (17) and Eqn. (18).

## Implementing a seq2seq model with attention as a cognitive model of human memory search

Having established the historical parallels and provided a detailed mathematical mapping between the seq2seq model with attention and CMR, we can now leverage their convergence to explore alternative ways to model human memory. For the proposed seq2seq model with attention, we use gated recurrent units (GRUs) due to their simplified structure[6]. The model's output/retrieval rule, as described in Eqn. (6), is specified to align closely with the retrieval rule in CMR, where an item's recall probability depends on the match between the current context and the item's study context (see Supplementary Methods section S1.1.). We begin by exploring the potential of the seq2seq model as a predictor of individual recall behavior and compare its explanation and predictive abilities to those of CMR. The analyses outlined in this study were not pre-registered. The analysis of secondary datasets in this study complied with all ethical regulations of Rutgers University.

**Explain and predict individual participant behavior.** For individual participant fitting, each participant's set of presentation-recall pairs was partitioned into training (90%), validation (5%), and test (5%) splits. Data splitting was performed randomly, ensuring that no presented list appeared in more than one subset. Seq2seq models with attention and a hidden dimension size of 128 were trained on the training set, with generalization loss evaluated on the validation set and final performance reported on the held-out test set. This individual fitting training was performed using a standard cross-entropy loss due to the order of recall being necessary to capture the recall characteristics of each participant. For this supervised training, each model is trained using the Adam optimizer with a learning rate of 0.001, $\beta_1 = 0.9$, and $\beta_2 = 0.999$. Training was performed using mini-batches of size 32 with an early stopping patience of 5 epochs, allowing each model to train for an unspecified number of epochs required until validation loss failed to improve for 5 epochs. No dropout or gradient clipping was used in training across any of the experiments. Training and validation loss curves indicated convergence of individual participant models. For comparison, a CMR model was fit using Bayesian optimization[49] for a total of 300 optimization iterations on each participant's training and validation splits. This method optimizes CMR parameters by minimizing the root-mean-square error between human and model-generated recall patterns (i.e., the serial position curve, the probability of first recall, and the conditional response probability). To ensure each pattern contributed equally to the optimization objective, we applied min-max scaling to normalize the RMSE of each curve type before aggregating. The final evaluation results of the seq2seq model and CMR model are shown for the test split with respect to each participant. Additional model details can be found in the Supplementary Methods section S1.1.

**Optimal free recall behavior.** In addition to evaluating the model's ability to explain and predict individual participant recall data, we train the model directly on the free recall task, rather than fitting to participant recalls, and examine its optimal recall behavior. The model training procedure is posed as a reinforcement learning problem, in which an agent is presented with a list of words, and then expected to recall as many items as possible from this list when prompted with a start-of-sequence token and terminate its own recall by the prediction of an end-of-sequence token. During the training, random lists are continuously generated for each episode, ensuring that the model is given exposure to every word appearing in the vocabulary. We use the proximal policy optimization (PPO) algorithm[50] to obtain the optimal policy. For the PPO algorithm, the entropy coefficient was set to 0.01, the discount factor was 0.99, and the PPO clip coefficient was 0.2. The reward structure during the model training is as follows: +1 for each correct recall, −1 for each incorrect recall, and −0.5 for repeating a previously correctly recalled word. Training consisted of 50,000 iterations, each performed on a batch of 4 episodes, where each episode represented the presentation

and recall of a single list. Additional model training details can be found in the Supplementary Methods section S1.2.

For evaluation, all model configurations are evaluated on 10,000 randomly generated sequences, and the resulting recalled lists are analyzed along the standard free recall metrics: the serial position curve, the probability of first recall, and the conditional response probability. To test the ability of our model to predict human data in patients with medial temporal lobe amnesia and healthy controls[36], we train two distinct configurations of the free recall model: the standard model as described and a model with the attention mechanism removed. In addition, we evaluate the effects of hidden dimension size, analogous to working memory capacity, by training and evaluating seq2seq models with different hidden dimension sizes: 32, 64, and 128.

**Penn electrophysiology of encoding and retrieval study.** For all model training and behavior comparisons with CMR and human participants, we use data from the Penn Electrophysiology of Encoding and Retrieval Study (PEERS)[51]. Our behavior comparisons focused on behavioral data from 171 young adults between the ages of 18 and 30 who completed Experiment 1 of the PEERS dataset, which consisted of an immediate free recall task. Information regarding gender and race could not be found in the original study. Each individual underwent one practice session followed by six experimental sessions, each containing 16-word lists. Only data from the experimental sessions were included in our training and evaluation data. Each word list consisted of 16 words and was immediately followed by a free recall test. Words were presented on-screen either with an associated encoding task—requiring a size judgment or an animacy judgment—or without any encoding task. The experiment included three types of word lists: no-task lists, single-task lists, and task-shift lists. In order to control for ordering effects, the sequence of lists and tasks was counterbalanced across sessions and participants. Words were displayed for 3000 milliseconds each, followed by a jittered inter-stimulus interval randomly selected between 800 and 1200 milliseconds. If a word was linked to an encoding task, participants responded by pressing a key. After the final word was presented and a jittered delay of 1200 to 1400 milliseconds elapsed, participants had 75 seconds to recall as many words as possible from the just-presented list.

**Medial temporal lobe amnesia and temporal context study.** For our ablation study of the attention mechanism, we compare the recall characteristics of our model without attention to those of patients with medial temporal lobe (MTL) amnesia[36]. In this experiment, the effect of MTL lesions on temporal context memory was studied using a looped-list free recall task design. This experiment sought to assess episodic memory impairments by presenting participants with 12-word lists, each containing ten high-frequency, high-concreteness nouns. These lists were repeated four times in a looped sequence to reinforce the temporal relationships between words. After each list, participants were asked to recall the words in any order immediately following the presentation. The study included ten patients with MTL damage and 16 healthy controls matched for age and education level. Information regarding gender and race could not be found in the original study. Each participant underwent three sessions, during which they were randomly assigned 9 of the 12-word lists, with the order of lists randomized both within and across sessions. During the task, each word was presented for 1200 ms, followed by a 1600 ms inter-trial interval, and participants were instructed to recall the words immediately after each list cycle. Their recall was recorded by the experimenter, noting the order, repetitions, and any intrusions (words not on the list).

### Reporting summary

Further information on research design is available in the Nature Portfolio Reporting Summary linked to this article.

## Results

The following section details analyses of the seq2seq model's behavioral patterns compared to those of CMR and human recalls. We begin by

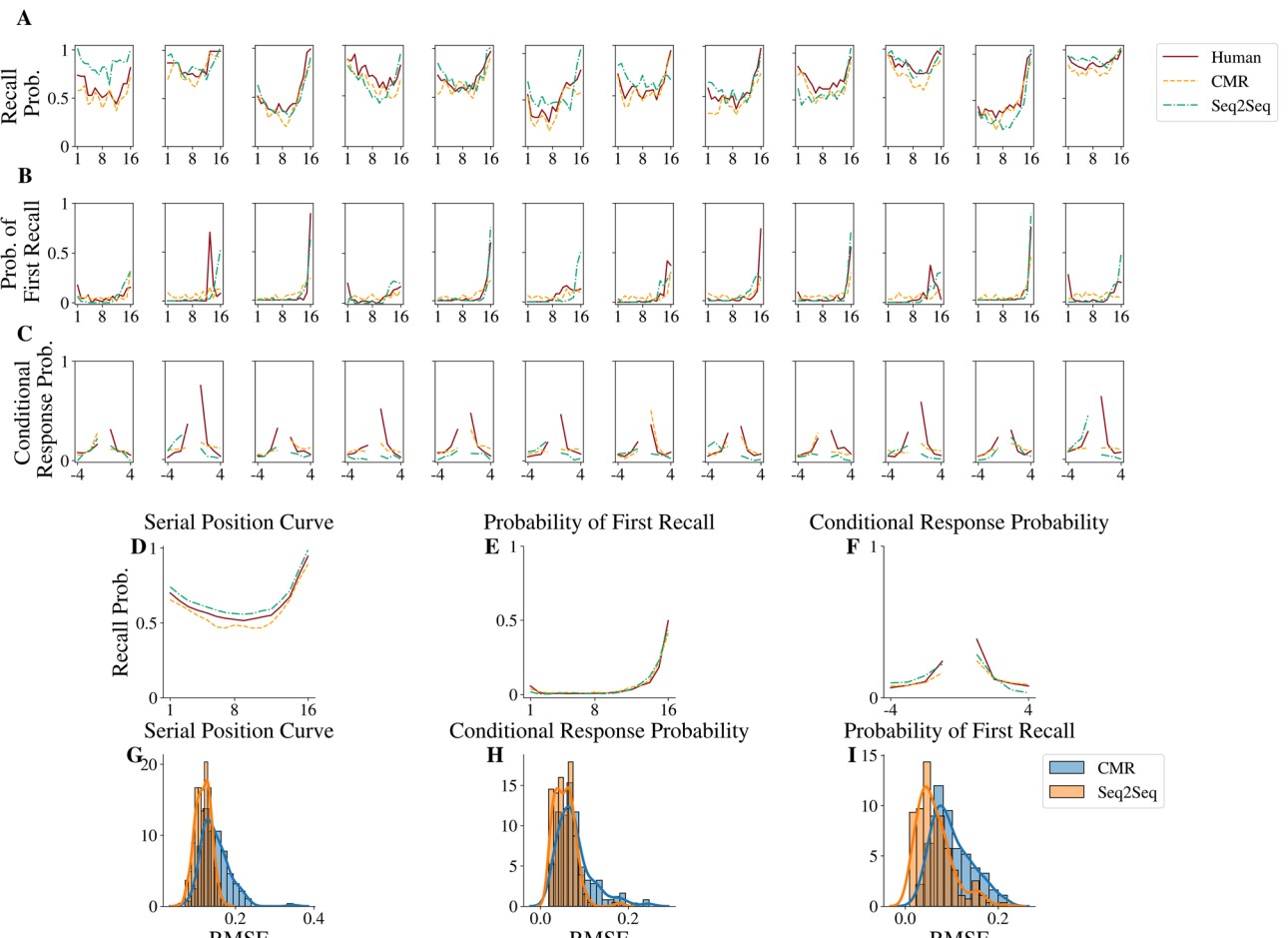

**Fig. 3 | Behavioral patterns for individual participants and the model predictions.**
**A–C** Behavioral patterns for the first 12 participants out of 171 participants, reproduced from Experiment 1 of the PEERS free recall dataset[51], overlaid with behavioral patterns of the CMR model and the seq2seq model with attention, trained over a separate subset of the same individual participant data. The behavioral patterns include (**A**) the serial position curve, (**B**) the probability of the first recall, and (**C**) the conditional response probability. **D–F** Aggregated behavioral patterns for all

171 participants overlaid with aggregated CMR and seq2seq model behavioral patterns. Model fits are quantitatively evaluated regarding the root-mean-square error between the human behavioral patterns of the 171 participants and model predictions over (**G**) the serial position curve, (**H**) the probability of the first recall, and (**I**) the conditional response probability. Both Seq2Seq and CMR model fitting analyses are for N = 171 participants.

analyzing the potential of these models to predict individual human recall behavior based on three sets of recall patterns: 1) how well on average the model or participant retrieves items for each position in the study list (the serial position curve), 2) where the model initiates its recall from (the probability of the first recall, and 3) how likely it is to recall items studied consecutively in the study list (the conditional response probability, computed by dividing the number of times a transition of that lag is actually made by the number of times it could have been made[33]). Next, we examine behavioral patterns of the fully optimized seq2seq models as well as behavioral patterns exhibited throughout the training process. Then, we explore the impact of changing hidden dimension size, drawing a parallel between the hidden dimension and working memory capacity in human participants. Finally, an ablation study is conducted to assess the impact of removing the attention mechanism, and we compare the resulting model deficits and memory impairments observed in medial temporal lobe patients with amnesia.

**The seq2seq model can explain and predict recall behavior in memory search comparably to the CMR model**
We begin by exploring the potential of the seq2seq model as a predictor of individual recall behavior and compare its explanation and predictive abilities to those of CMR. In order to assess the predictive ability of the two models, both CMR and the seq2seq model (128-Dim. with attention mechanism)

were fit to a training split of each participant's recall data (rather than aggregated data across all participants) and then evaluated on a hold-out test set from the same participant. For the CMR model, fitting is accomplished by optimizing CMR parameters using Bayesian optimization to minimize the root-mean-square error between model-generated and human behavioral patterns. The seq2seq model is fit to human data using human participant-presented lists as model input and human participant recalled lists as ground-truth with a standard cross-entropy loss as the objective function[52]. Figure 3A–C depict the behavior patterns of the first 12 participants (for illustrative purposes) of the 171 participants taken from the PEERS dataset with both fitted CMR and seq2seq models overlaid. Additionally, we show the average recall behavioral patterns for all 171 participants in Fig. 3D–F. While both CMR and the seq2seq model have qualitatively good fits to the human recall data, we compare their model fit quality in terms of the root-mean square error between the human recall behavior and corresponding model recall behavior across all 171 participants (Fig. 3G–I). We performed a series of two-sided Wilcoxon signed-rank tests to compare each pair of distributions for the root-mean-square-error fit for both CMR and the seq2seq model. We find that the seq2seq model fits the corresponding human data curve with significantly smaller error for the serial position effects (Wilcoxon signed-rank test: two-sided, $W = 0.0$, $n = 171$, $p = 8.2 \times 10^{-30}$, ranked biserial correlation = −0.995), the probability of first recall (Wilcoxon signed-rank test: two-sided, $W = 2880.0$, $n = 171$, $p = 5.23 \times 10^{-12}$, ranked biserial

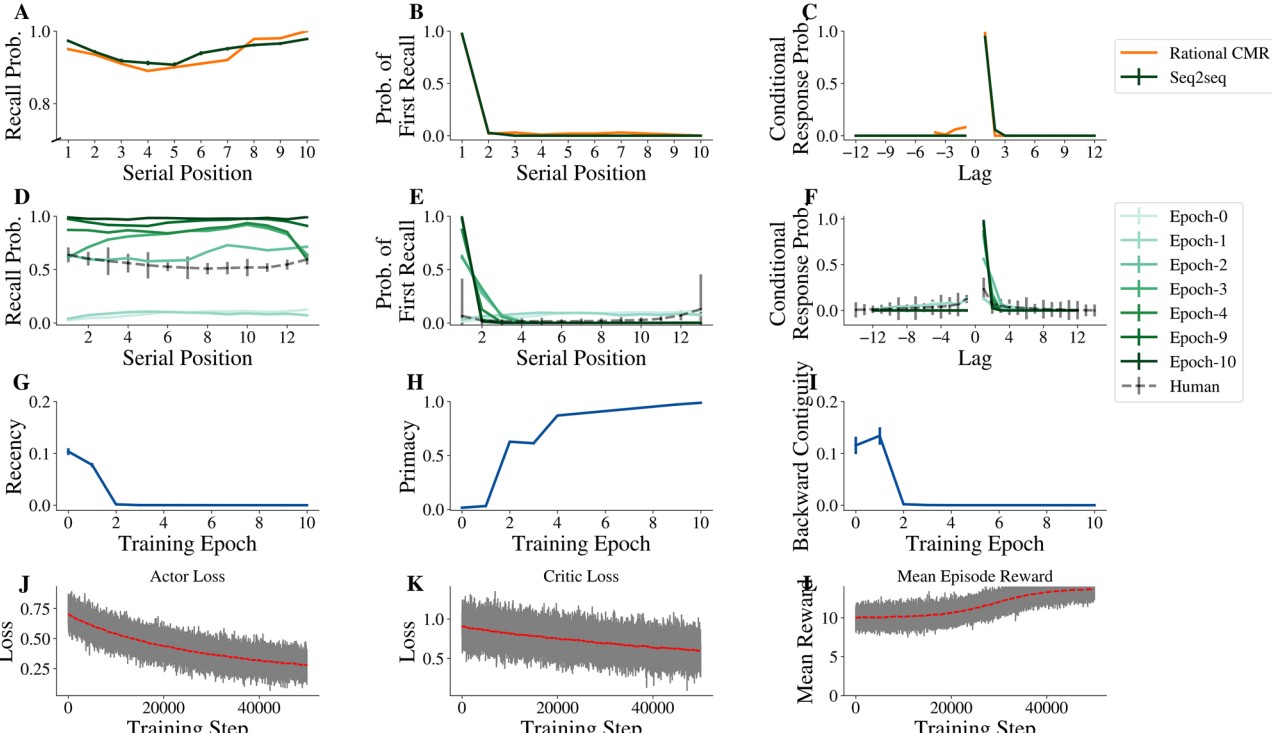

**Fig. 4 | Optimized and intermediate training of the Seq2Seq model with attention.** The fully optimized seq2seq model with attention (128-Dim) shows behavioral patterns that closely align with those of the rational-CMR model across three sets of free recall patterns: serial position curve (**A**), probability of first recall (**B**), and conditional response probability (**C**). Rational-CMR results are reproduced from Zhang et al.[31]. **D–F** Recall characteristics are evaluated for intermediate epochs of training and compared qualitatively to the average participant recall from the PEERS dataset (N = 171). Intermediate model results exhibit recency effects and backward contiguity effects that are typical in human behavioral patterns. **G–I** The same results are plotted along the dimension of training epochs, summarizing the model's tendency to initiate recall from the end (the last three items of the list; **G**), to initiate recall from the beginning of the list (the first three items of the list; **H**), and to recall items in the backward direction (conditional response probability with −1 lag; **I**). **J–L** Actor and critic loss curves exhibit a consistent decrease over time, and the mean episode reward converges toward the maximum possible recall reward of 14 for our experiments. 5000 training steps correspond to 1 training epoch.

correlation = − 0.608), and the conditional response probability (Wilcoxon signed-rank test: two-sided, $W = 3822.0$, $n = 171$, $p = 5.51 \times 10^{-8}$, ranked biserial correlation = −0.480). These results are not affected by different methods of conducting the training and testing splits (see Supplementary Discussion S2.2). When the amount of training data is reduced, however, CMR is shown to give better predictions, highlighting a trade-off between model flexibility and data efficiency (see Supplementary Discussion S2.3). Together, these results support that the seq2seq model with attention can capture human recall patterns as effectively as CMR. This ability to capture human recall patterns is not a result of the neural network model being over-parameterized, as the trained model was able to predict behavioral patterns in the unseen portion of the recall data.

### The optimized seq2seq model demonstrates the same recall behavior as the optimal policy of the CMR model

Past cognitive modeling work with CMR has demonstrated that not only could it capture averaged human behavioral patterns (through fitting model parameters to human data[19,20]), but it can explain why some individuals achieve better memory performance than others in the free recall task by analyzing how well their behavior aligns with an optimal policy of CMR (rational-CMR[31]). When the free recall behavior is optimized under the architectural constraints of the CMR model, the optimal policy begins by recalling from the beginning of the list and sequentially recalling forwards despite no constraint placed on the order of recall (Fig. 4B). Intuitively, recalling items in a list in the same order they are studied helps minimize the odds that an item will be inadvertently skipped. However, this optimal behavior is non-trivial because the end-of-list context is readily available at the start of recall, whereas reactivating the beginning-of-list context necessitates a potentially costly context shift to early items of the list. Why is

it necessary to start recalling from the beginning and recall forward, rather than from the end and then recall backward? In CMR, each item is encoded along a gradually drifting internal context. During recall, retrieving an item brings back the context from when that item was originally studied, which then serves to cue recall of other items. During this process, there is a reliable way to proceed forward (by reinstating the pre-experimental context associated with the item), but there is not a reliable way to proceed backward —the only way to promote backward transitions is to reinstate the experimental context associated with the item at study, which is equally likely to propel recall forward and backward. As a result, it yields better performance when initiating the recall from the beginning of the list and then recalling forwards[31]. Figure 4A–C shows that, when our seq2seq model with attention (dark green) is trained to optimize recall performance using reinforcement learning, it demonstrated the same behavior as the optimal policy of CMR (orange; Fig. 4A–C reproduced from Zhang et al.[31]). We additionally plot the actor and critic losses along with the mean episode reward across training (Fig. 4J–L) to demonstrate convergence of the seq2seq model with attention. Both models achieve near-optimal performance (Fig. 4A), with a near certainty of starting recall at the beginning of the sequence (Fig. 4B) and making a forward recall transition to the next serial position at lag +1 (Fig. 4C). These simulations provide evidence that the seq2seq model with attention shares similar architectural constraints as those of CMR, giving rise to optimal behavior aligned with that of the optimized CMR.

### The intermediate training evaluations of the seq2seq model exhibit similar qualitative patterns as are typically observed in human participants

Only a small proportion of top-performing human participants can demonstrate the exact behavior of the optimal policy[31]. Averaged human

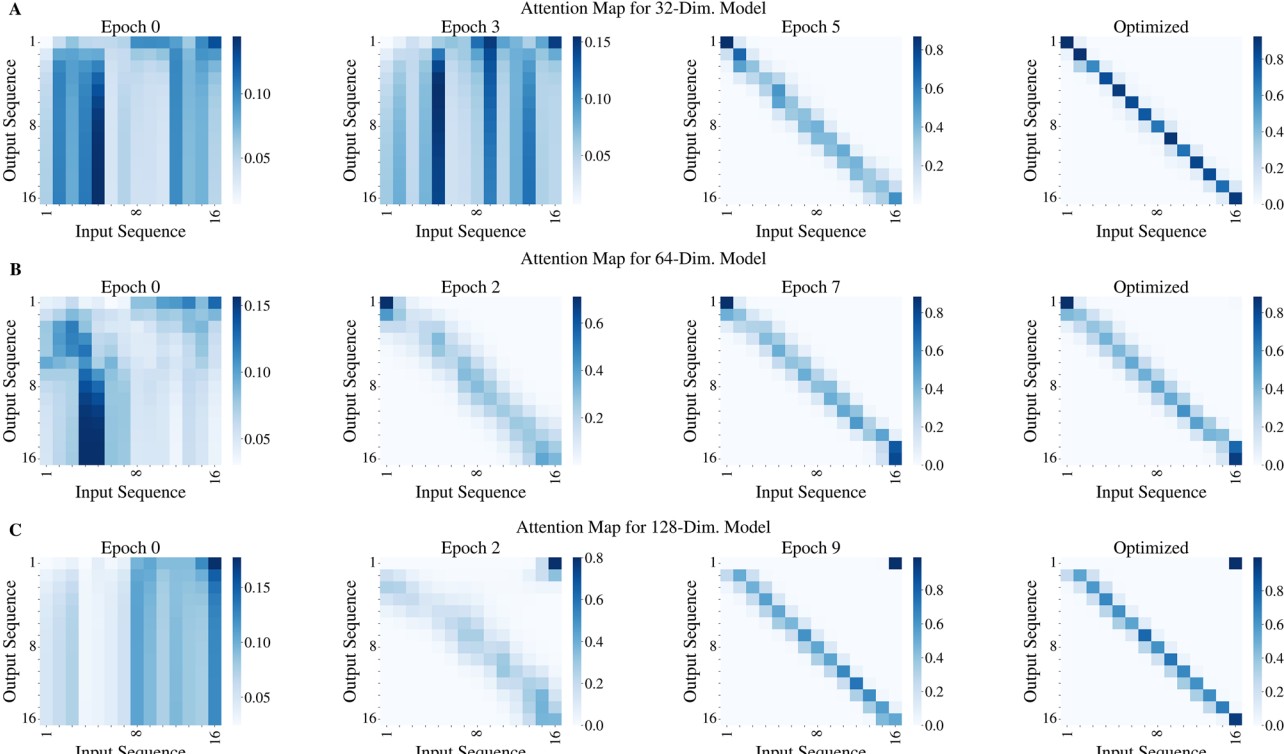

**Fig. 5 | The effect of working memory capacity (hidden dimension size) on optimal recall behavior. A–C** Heat maps illustrating the evolution of attention weights for seq2seq models with varying hidden dimension sizes across training epochs. Attention weights are averaged across all trials and indicate the influence of each encoded item (y-axis) on each decoding step (x-axis) during recall. **A** With a small hidden dimension size (32-Dim.), attention shifts over epochs toward the beginning of the input sequence, supporting the primacy effect through attention alone and enabling sequential forward recall from the beginning of the list.

**B** Attention results with 64-Dim. hidden dimension size show similar results to those of the 32-Dim. model, but more quickly approaching sequential forward recall and showing weaker attention weights in the fully optimized stage. **C** With a large hidden dimension size (128-Dim.), attention for the first item consistently favors the last encoded item, suggesting that primacy is achieved not via attention shifts but by retaining the list start within the model's hidden state (analogous to working memory).

recall behavior in free recall experiments (dashed line in Fig. 4D–F; reproduced from Kahana et al.[51]) also exhibits recency (enhanced end-of-list recall) and backward contiguity (tendency toward shorter than longer backward lags), in addition to what is amplified in the optimal behavior with primacy (enhanced beginning-of-list recall) and forward contiguity (tendency toward shorter than longer forward lags). To assess sub-optimal model behavior, we evaluate the recall behavior of several intermediate checkpoints taken of the model throughout the optimization process (Fig. 4D–F). Characterizing these intermediate training stages provides insight into how recall strategies emerge and change as the model learns. We show that while the fully optimized seq2seq model with attention demonstrates the same behavior as the optimal policy of CMR, exhibiting both primacy and forward contiguity (Fig. 4A–C), intermediate training evaluations of the model (lighter green lines) exhibit recency (higher recall probability at the end of list in Fig. 4D in early Epochs 0–2) and backward contiguity (positive transition probabilities at negative lags in Fig. 4F) similar to those observed in averaged human recall behavior. As training progresses, the model quickly loses recency and backward contiguity, increasingly depending on forward recalls that initiate recall from the beginning of the sequence. This trend is made more evident in Fig. 4G–I, which displays the change in the model's tendency to initiate recall from the end (the last three items of the list; Fig. 4G), to initiate recall from the beginning of the list (the first three items of the list; Fig. 4H), and to recall items in the backward direction (conditional response probability with -1 lag; Fig. 4I). Although model behavior during intermediate training evaluations is partially influenced by model initialization, we find that patterns such as recency and backward contiguity reliably appear across training runs and initializations, suggesting that these are stable features rather than idiosyncratic fluctuations.

## The seq2seq model provides insight into how working memory capacity affects optimal recall strategies

While the findings above support seq2seq models as alternative models of human free recall, for the remaining analyses in the Results section, we conduct modeling analyses to highlight the additional strengths of the seq2seq model. Critical to the recall performance for both CMR and the seq2seq model with attention is the ability to initiate recall from the beginning of the list, i.e., the primacy effect. While CMR allows direct access to the beginning of the list during recall without an explicit mechanism, the seq2seq model with attention model provides additional insight into how the primacy effect emerges during learning. To illuminate the underlying learning process, we analyzed how the attention mechanism evolves across training epochs. The attention weights represent the relative importance placed on each encoding context at each step of the decoding process, as defined in Eqn (3). These weights are visualized at various stages of training as a heat map (Fig. 5) representing the attention weight between each decoder hidden state (x-axis) and encoder hidden state (y-axis), averaged across all training trials. Figure 5 depicts these averaged heat maps for models with different hidden dimension sizes. When the hidden dimension is small (32 in Fig. 5A), primacy is achieved through the attention mechanism itself. The attention heat-map at Epoch-0 demonstrates that at the beginning of the recall, only the context of the later list items of the input sequence is readily available, as attention is distributed across the second half of the input sequence before recalling the first item of the output sequence. With each successive epoch of optimization, the attention heat-map shifts toward the optimal behavior, where most attention is placed on the first item of the input sequence at the beginning of recall (Fig. 5A, Epoch 5 and Optimized). Recalling the first item in the input sequence subsequently

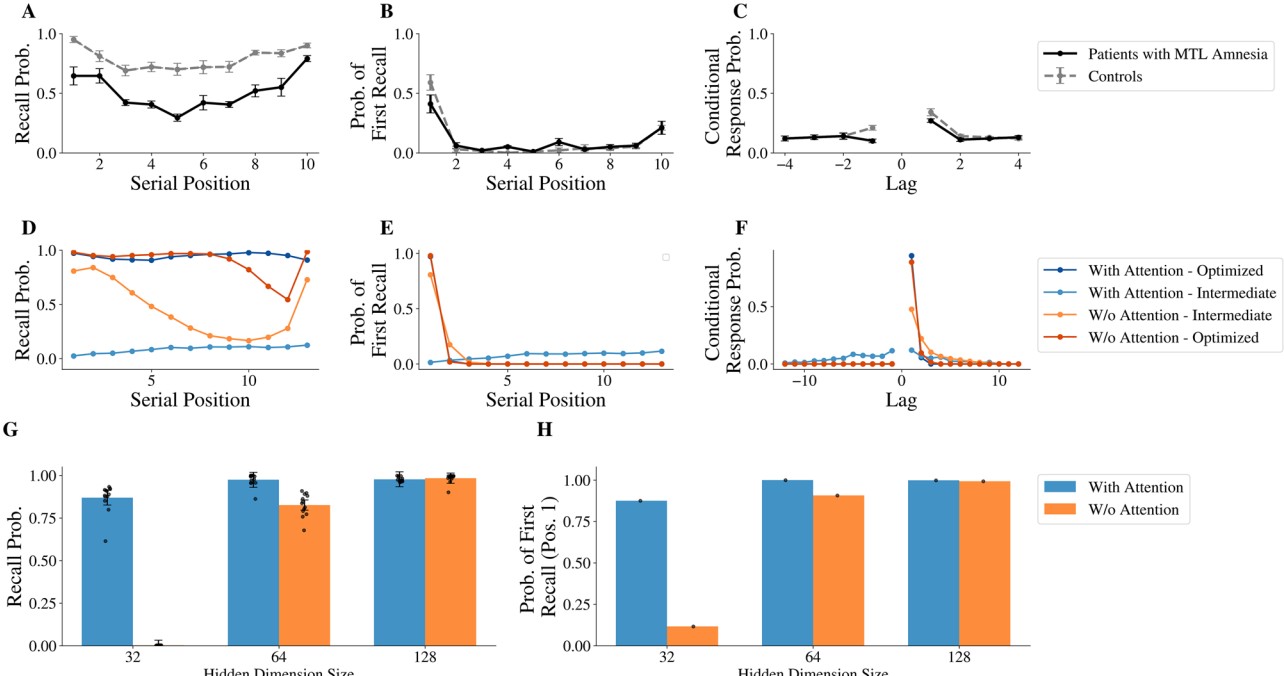

**Fig. 6 | Comparing optimized model behavior under ablation of the attention mechanism with recall behavior of medial temporal lobe (MTL) patients with amnesia. A–C** Behavioral patterns of patients with MTL amnesia ($N = 10$) compared to healthy controls ($N = 16$), demonstrating memory deficits on a free recall task, reproduced from Palombo et al.[36]. Despite the small list size, patients with MTL amnesia exhibit diminished recall performance and reduced ability to perform backward recall transitions (conditional response probability at −1 lag). **D–F** Behavioral patterns of the seq2seq model with and without attention (hidden dimension size 64), optimized or during intermediate training. The model without attention exhibits diminished recall performance and reduced ability to perform backward recall transitions (no backward contiguity at any stage of training as seen from the conditional response probability at −1 lag). **G, H** Behavioral patterns of models of varying hidden dimension sizes with and without attention. Models without attention can achieve the optimal recall behavior but require sufficiently large hidden dimensions. ($N = 13$ for each dimension size evaluation).

activates the attention on the next item in the input sequence, forming a diagonal pattern that characterizes recalling items sequentially forward.

Next, we use differences in hidden dimension size to model differences in working memory capacity. This view of working memory is consistent with previous modeling work in which recurrent neural activity is used to model active maintenance and integration of information[2,53] and work in which capacity is affected by the number of hidden units in the recurrent neural networks[54]. Larger hidden dimension sizes allow more information to be held in the hidden state, though this only provides a relative measure of working memory—hidden dimension size does not directly translate to a specific number of items that can be held in working memory in human participants. We hypothesize that an increase in the model's working memory capacity (as modeled by hidden dimension size) reduces the model's need to obtain primacy through the attention mechanism. Indeed, we observed that when the hidden dimension is large (128 in Fig. 5C), only the last item of the input sequence is available at the beginning of recall, and this pattern does not change even after epochs of optimization (Fig. 5C, Optimized). The observation that the model with a hidden dimension size of 128 can still initiate recall from the beginning of the input sequence indicates that primacy is not obtained through the attention mechanism but through maintaining the item in the hidden state itself, enabled by a larger working memory capacity. While there is alignment in the optimal behavior of CMR and the seq2seq model with attention, visualizing attention weights across intermediate stages of training in the seq2seq model with attention offers additional insights into how the optimal behavior is achieved.

### An ablation study of the attention mechanism provides insight into the performance difference between patients with amnesia and healthy controls

Medial temporal lobe (MTL) lesions have been previously associated with the inability to reinstate prior experienced contexts[35–38], mechanistically

corresponding to the context reinstatement mechanism in CMR. Figure 6A–C depict recall patterns for patients with MTL amnesia and healthy controls, reproduced from Palombo et al[36]. In addition to the difference in recall probability between patients with amnesia and healthy controls (Fig. 6A), patients with amnesia also lack the ability to "jump back in time" and therefore display a greatly reduced backward contiguity relative to healthy controls (lower −1 lag in 6C), consistent with CMR's predictions when the ability to reinstate the original study contexts is impaired. As we hypothesized that the attention mechanism in our seq2seq model is analogous to context reinstatement in CMR, we carried out an ablation study of the attention module to examine if models with and without attention, when trained to optimize recall performance, could capture the key behavioral differences in the healthy controls and patients with amnesia, respectively. Results of this ablation are displayed in Fig. 6D–F, alongside the model with attention, where both models are trained with a hidden dimension size of 64. We found similar results with the human data in the recall performance, where the attention model can recall more information than the no-attention model after being fully optimized (Fig. 6D). Moreover, the lack of the attention mechanism in Fig. 6F largely eliminates the model's capacity for backward contiguity (at any stage of training) compared to the attention model, where backward contiguity is observed in early iterations of training. Additional model simulations (see Supplementary Discussion S2.1) provide evidence that these behavioral differences are due to the presence of the attention mechanism rather than simply due to the attention model having more parameters. These ablation results support the idea that the attention mechanism in the seq2seq model is analogous to the context reinstatement mechanism in CMR (which has previously been linked with MTL damage[36]) and contribute to memory performance during free recall. Importantly, the differences in recall performance and backward contiguity between attention and no-attention models do not come from direct fitting to human data but emerge during the process of optimizing performance, revealing the

functional role of the attention mechanism in contributing to memory performance.

To further understand the role of the attention mechanism in contributing to the performance difference between healthy controls and patients with amnesia, we examine the effect of hidden dimension size. Figure 6G, H show evaluation results for each configuration with respect to hidden dimension size and the presence of the attention mechanism after full optimization. Figure 6G shows that the attention model exhibits higher recall probability than the no-attention model when the models' hidden dimensions are small (32 Dim: Wilcoxon signed-rank test, $W = 91.0$, $n = 13$, $p = 1.22 \times 10^{-4}$, ranked biserial correlation = 0.986, one-sided; 64 Dim: $W = 91.0$, $n = 13$, $p = 1.22 \times 10^{-4}$, ranked biserial correlation = 0.993, one-sided), but not when the hidden dimension size is large (128 Dim: $W = 29.0$, $n = 13$, $p = 0.878$, ranked biserial correlation = $-0.363$, one-sided). At smaller hidden dimension sizes 32 and 64, the ablation of the attention mechanism captures recall deficits observed in patients with amnesia compared with healthy controls, shown in Fig. 6A. Since the fully optimized versions of both the attention model and the no-attention model demonstrate optimal free recall behavior (initiating recall from the beginning of the list and recalling sequentially forwards; Fig. 6E–F), one might ask why the attention model has better memory performance when it must learn to ignore backward contiguity, while the no-attention model has forward contiguity by default (i.e., incapable of backward contiguity). Evidence from the optimal CMR model shows that forward contiguity enhances performance only when coupled with primacy[31]. We hypothesize that the lack of the attention mechanism (i.e., the ability to reinstate previous study contexts) makes it difficult to reinstate the beginning-of-list context, which is necessary to establish primacy. We argue that the absence of primacy leads to overall diminished performance in the no-attention condition. An alternative, and more difficult way, to obtain primacy in the no-attention model is to maintain the first item in the original sequence in the working memory (i.e., hidden states), and this difficulty is greater when the working memory capacity is smaller. This greater difficulty was made evident by the stronger reliance on the attention mechanism by the smaller hidden dimension sizes, while the largest, 128-Dim model was able to maintain the first item without the use of the attention mechanism (Fig. 5). Aligned with our prediction, the gap in performance between the attention-based and no-attention models was larger when the models had smaller hidden dimension sizes. A hidden dimension size of 32 is inadequate to observe any recall performance in the no-attention case (Fig. 6G), which is associated with a lower primacy value compared with the model with attention (Fig. 6H). This is in contrast to a hidden dimension size of 128 where models with and without attention demonstrate similar memory performance (Fig. 6G), as one could obtain primacy either through maintaining the first item in the working memory without requiring the attention mechanism, as seen from similar primacy values in Fig. 6H.

## Discussion

We have demonstrated the close correspondence between neural machine translation (NMT) models and cognitive models of human memory by providing a detailed mathematical mapping between architecture components in the sequence-to-sequence (seq2seq) model with attention to those in the CMR model of free recall. We showed that seq2seq models with attention mechanisms, originally introduced for the purpose of machine translation, can serve as a neural network model of human memory search. Several seq2seq model components align closely with mechanisms in the CMR model, and the addition of attention provides the model with a form of mental time travel, enabling explicit reactivation of prior contexts much like CMR. By fitting to human recall data, we show that the seq2seq model can predict human recall characteristics with higher accuracy than the CMR model, indicating its viability as a cognitive model of human free recall. Comparisons to the optimal policy of CMR illustrate the model's alignment with CMR in optimal recall behavior, while comparisons between human participant data and incompletely optimized models reflect the model's potential for capturing sub-optimal recall behavior as well. In addition to

establishing this seq2seq model as an alternative memory search model (capable of predicting individual participant behavior and capturing optimal and sub-optimal recall behavior as CMR does), we demonstrated additional strengths of the seq2seq model in evaluating the role of context as maintained in the working memory versus that reactivated from the episodic memory. We showed that reduced working memory capacity (as hidden state dimension sizes are altered) requires a compensatory mechanism and a stronger reliance on episodic memory to support optimal recall behavior. By removing the attention mechanism and hampering the model's ability to reinstate previous encoding contexts, we eliminate the model's capacity for backward contiguity and reduce the model's recall performance in a manner similar to patients with MTL amnesia. We now turn to the broader implications of these results.

### An alternative model of human memory search

Our present work uncovers parallels between model architectures of neural machine translation and model architectures of human memory search, which opens up opportunities for improving existing models of human memory. The seq2seq model with attention strikes a unique balance between interpretability and flexibility. This balance is crucial for cognitive modeling, where its goal lies in both the transparency in arriving at its outputs and the ability to capture complex human behavioral data. In comparison, purely data-driven machine learning methods can often outperform cognitive theories in capturing and predicting human behavior[55–57] and are sometimes considered as an upper bound on the amount of variance we can expect to account for in the human data[58–61]. However, it can be challenging to interpret off-the-shelf machine learning methods[62]. The seq2seq models with attention do not suffer from the challenge of interpretability, as we have demonstrated in this work how each component – such as the attention mechanism or the updating of hidden states—can be mapped mathematically onto components of CMR. In addition to being interpretable, the model remains sufficiently flexible to capture a wide range of complex behaviors observed in human free recall. Our analysis reveals that the seq2seq model with attention serves as an effective substitute for the CMR model in fitting and predicting individual free recall characteristics. The seq2seq model not only aligns closely with human recall patterns but also outperforms CMR in certain predictive tasks. This supports the idea that the seq2seq approach could be adopted as a useful tool in cognitive modeling for understanding human memory search. While the current work focuses on RNN-based models, Large Language Models (LLMs) offer another promising avenue for modeling human memory. There is evidence that pre-trained LLMs exhibit striking similarities to human episodic recall, including serial position effects[63,64] and temporal contiguity[65,66], without being explicitly trained on memory tasks. Recent work suggests some of these behaviors, particularly the temporal contiguity effects, emerge from the activity of induction heads within the transformers, which have been directly mapped to mechanisms in cognitive models of human memory like the Context Maintenance and Reinstatement (CMR) model[65]. Additionally, ablating these specific heads eliminates the structured recall patterns[66]. Identifying convergences between human memory and machine learning models not only deepens understanding of human memory but also drives improvements in machine learning algorithms. In one such study, Fountas et al.[67] improve the LLM's ability to effectively use its context by explicitly integrating key aspects of human episodic memory and event cognition into the model architecture.

### Disentangling the contributions of working memory and episodic memory

Recalled items during a free recall task, as specified by the CMR model, are driven by the current context, which is a combination of an evolving context from the previous time step (as maintained from the working memory) and a context reactivated from the encoding period (as retrieved from episodic memory, analogous to the attention mechanism). The CMR model can determine the combination of the two contexts in a single and fixed parameter, but cannot derive the dynamic interaction of these two contexts over

time and their relation to free recall performance. Unlike CMR, the seq2seq model has in place of this single fixed parameter a learned dense layer that can dynamically combine the two contexts in order to drive recall performance. This process helps disentangle the contributions of two types of contexts and identify their changes over each time step of the recall sequence (Fig. 5). We showed that seq2seq models with smaller hidden dimensions, analogous to a limited working memory capacity, exhibit different recall strategies and a stronger reliance on the attention mechanism to access primacy items compared to models with larger hidden dimensions. Our results indicate that the need to resort to episodic memory only arises when other mechanisms, such as those that support active maintenance, are exhausted.

These results are consistent with previous neuroscience literature showing that after participants studied a short list of items, access to primacy items was not accompanied by activation of the medial temporal lobe[68]. In contrast, the medial temporal lobe is involved in studies where items were unfamiliar, complex, or involved relational processing, placing high demands on working memory[69,70]. In a similar vein, it has been proposed that if the material to be learned exceeds working memory capacity or if attention is diverted, performance depends on episodic memory retrieval even when the retention interval is brief[71]. In other words, MTL lesions would impair performance only when working memory is insufficient to support performance. Our model simulations support this interpretation by demonstrating that with ablation of the attention mechanism, our models demonstrate similar behavioral recall deficiencies as those in hippocampal patients with amnesia, but these deficiencies disappear if working memory capacity (hidden dimension size) is increased.

### A rational explanation of architectural assumptions in human memory search

Cognitive models comprise a set of computations and mathematical equations that implement theoretical accounts of cognitive processes[72]. A good fit of the cognitive model to human behavioral data reflects an accurate psychological theory, whereas a bad fit would refute it. Although such a model-fitting procedure has been a common practice in cognitive modeling and is useful in evaluating alternative psychological theories, we still do not know where a particular cognitive process comes from in the first place. Why do people tend to remember early-list items better than middle-list items (i.e., the primacy effect)? Why do people demonstrate a bias in forward transitions during free recall (i.e., forward asymmetry)? An alternative approach in cognitive modeling that can answer these questions is to build models that optimally solve the problem of a cognitive task[29,30,73,74]. If a set of memory behavioral patterns arises as a result of optimizing the task performance, then we can better understand the functional purposes of the cognitive processes underlying these patterns[2,31,75,76]. For example, our current work based on seq2seq models provided a rational account of why we observe primacy and forward asymmetry (as they contribute to achieving good task performance in free recall). We showed that initiating recall from the beginning of the list and then sequentially recalling in the forward direction gives rise to optimal recall performances, reproducing the same results as the optimal behavior derived based on CMR[31].

Critical to this rational approach is the specification of basic architectural assumptions about internal representations and processes during memory search[74,77,78]. These include how context evolves towards each item during encoding and is used to drive recalls, as specified in both the seq2seq model and the CMR model. Optimizing performance without these assumptions, such as allowing free traversal to any position in the context space regardless of previous context, can easily achieve the goals in memory search, but does not provide a realistic model of how human memory functions. While a rational approach can provide answers to why certain cognitive processes are important, we do not yet know if there is any adaptive purpose the architectural assumptions themselves serve or if they are merely hard constraints that limit the performance of the cognitive task. A similar procedure to address this question is to iterate through all possible architectures and examine which architecture out of this vast space of architectures produces optimal task performance. Although it is unrealistic

to carry out such an analysis in a single project, we argue that the field of neural machine translation has effectively conducted this analysis as researchers have explored countless architectures in search of those that optimize task performance. The convergence between the development of human memory search models (driven by alignment with human behavioral data) and the development of neural machine translation models (driven by task performance) provides evidence that the architectural assumptions specified in CMR serve an adaptive purpose.

### Limitations

One limitation of our current study is that we evaluate our model's ability to predict recall patterns over one type of free recall task when participants are asked to recall immediately after the encoding phase. There are other variants of the free recall task, where participants are required to recall the list after a delay period[79] - introducing a context change between encoding and recall, or where there are distractor tasks in between every pair of adjacent words during encoding[80] - introducing a context change between encoding adjacent words. CMR can capture these tasks through the amount of context drift. We expect that the seq2seq model can similarly account for these tasks, given the mapping we have derived between the seq2seq model architecture and the context drifting mechanism in CMR, and the flexibility in the seq2seq model to learn from the complex dynamics of context change. It remains a fruitful venue in the future to evaluate the seq2seq model to account for memory phenomena in different variants of the free recall task. Another limitation in the current work is our focus on examining the relationship between the seq2seq model with one specific memory search model, CMR. In addition to the CMR model, there is another prominent model of memory search, the Search of Associative Memory (SAM) model[81], which relies on the distinction between a short-term store and a long-term store without reactivating prior contexts at retrieval. Both SAM and CMR have a history of successfully accounting for various behavioral patterns in memory search. Despite similar levels of empirical evidence they receive, the convergence of CMR and the seq2seq model, as shown in the present work, provides additional support to the CMR model in terms of its computational efficiency and adaptability.

### Conclusions and future directions

In conclusion, we have demonstrated a convergence between neural machine translation models and cognitive models of human memory. Establishing a parallel between these models helps illuminate the factors behind the success of neural machine translation and opens up the opportunity to build more flexible models of human memory search. The seq2seq model with attention, with its balance of interpretability and flexibility, emerges as a useful tool for both simulating cognitive phenomena and exploring additional theoretical insights into memory processes. Compared with existing models of memory search that were built upon well-controlled stimuli like lists of random words, the seq2seq model with attention, traditionally applied to machine translation tasks, is suitable for handling more complex input and output structures such as sentences. An exciting venue for future research is to extend the application of the seq2seq model of free recall from recalling lists of words to recalling more naturalistic stimuli like narratives and stories. Future work could also expand on current findings by exploring the application of more advanced neural machine translation architectures, such as transformer-based architectures, in cognitive modeling and by further investigating the relationship between neural network architecture and human cognitive capacities.

### Data availability

The PEERS dataset[51] used to train and evaluate human behavioral fits is publicly available at PEERS Dataset.

### Code availability

The code used for training and evaluating model behavior is available at Code Repository. This repository contains all the necessary scripts and instructions to replicate the experiments described in this study.

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

## Acknowledgements

We would like to thank Marc Howard, Zoran Tiganj, and Qihong Lu for the helpful discussions. A subset of the results has appeared in a conference paper in the non-archival conference proceedings of the Annual Meeting of the Cognitive Science Society in 2024. This work was supported by a start-up fund from Rutgers University—New Brunswick and a grant from the National Science Foundation (2316716) awarded to Q.Z. The funders had no role in study design, data collection and analysis, decision to publish, or preparation of the manuscript.

## Author contributions

N.S. contributed to the conceptualization, data curation, formal analysis, investigation, methodology, software, validation, visualization, writing. Q.Z. contributed to the conceptualization, investigation, methodology, supervision, funding acquisition, and writing.

## Competing interests

The authors declare no competing interests.
