## [Transparent Peer Review file · Communications Psychology]

Sequence-to-Sequence Models with Attention Mechanistically Map to the Architecture of Human Memory Search

Corresponding Author: Dr Qiong Zhang

Version 0:

Decision Letter:

Dear Dr Zhang,

Thank you for your patience during the peer-review process. Your manuscript titled "Leveraging Convergence of Machine Learning and Context-Based Human Memory Models to Improve Modeling of Human Memory Search" has now been seen by 3 reviewers, and I include their comments at the end of this message. They find your work of interest but raised some important points. We are interested in the possibility of publishing your study in Communications Psychology, but would like to consider your responses to these concerns and assess a revised manuscript before we make a final decision on publication.

As agreed via email, we have consulted with colleagues from the Human Behaviour and the Neuroscience teams in Nature Communications on whether they would invite a transfer of the (revised) manuscript for further consideration in the journal. I regret to tell you that following their own evaluation of the manuscript and reviewers' reports, they declined a transfer of the work to Nature Communications.

We invite you to revise and resubmit your manuscript to Communications Psychology, along with a point-by-point response to the reviewers. Please highlight all changes in the manuscript text file.

Editorially, we consider it crucial that the central claims of the study, such as the superior performance of seq2seq model over CMR and that the attention mechanism in the seq2seq model captures behaviour in amnesia patients are quantitatively well supported. Additionally, please provide more details in the training and testing of the models, such as the exact loss function and parameter used, as well as how cross-validation is performed.

I am attaching an Editorial Requests Table that details critical reporting requirements for the revised manuscript. Please attend to each item and ensure your manuscript is fully compliant. If your revised manuscript is not aligned with these requests on major issues, such as those concerning statistics, it may be returned to you for further revisions without re-review.

Please submit the following items:

- Revised manuscript
- Point-by-point response to the referees' comments
- Cover letter (as a separate document)

- <https://www.nature.com/documents/nr-reporting-summary.zip>>Nature Research Reporting Summary
- <https://www.nature.com/documents/nr-editorial-policy-checklist.pdf>>Editorial Policy Checklist
- Completed Editorial Request Table (attached).

via this link: Link Redacted .

Additional guidance is available in our style and formatting guide Communications Psychology formatting guide.

Best regards,

Troby Lui

Troby Lui, PhD
Associate Editor
Communications Psychology

REVIEWER EXPERTISE:

Reviewer #1: Memory (CMR)
Reviewer #2: Memory (CMR)
Reviewer #3: Machine learning

REVIEWER REPORTS:

Reviewer #1 (Remarks to the Author):

This work explores the parallels between neural machine translation (NMT) models, specifically sequence-to-sequence (seq2seq) models with attention mechanisms, and context-based human memory models, specifically the Context Maintenance and Retrieval (CMR) model. The authors claim that these two model types, despite being developed for different purposes (task performance optimization in NMT and alignment with human behavioral data in cognitive science) exhibit strong architectural similarities. By mapping components like the seq2seq encoder-decoder structure and attention mechanism to CMR's context drift and reinstatement processes, they demonstrate that the seq2seq model can effectively serve as a cognitive model of human memory retrieval, specifically in free recall. The paper claims that this convergence validates the adaptive role of context in human memory and also introduces a flexible, interpretable neural network alternative to traditional cognitive models, capable of capturing both averaged and optimal human recall patterns as seen in free recall tasks.

Overall, this approach provides a novel and insightful parallel between cognitive modeling and neural networks.

While the study is already comprehensive and detailed, I have some suggestions to potentially improve clarity and interpretability

Eq. 4 might contain an error - it states that both item to context and context to item matrices are identical. Shouldn't item to context association be $f_{tc_{t-1}}^T$ and context to item $c_{t-1}f_t^T$. More explicit mention of dimensions for each of the vectors and matrices would also help with understanding.

Please provide more information about the neural network and the training process: number of parameters, learning rate, optimizer.

Is there anything specific in the attention computations - would you expect different results if the attention layer was replaced with dense?

The statistical analysis on page 8 leads to a conclusion that "seq2seq model fits the corresponding human data curve with significantly smaller error for serial position, probability of first recall and conditional response probability". This is not particularly apparent from the data from 12 selected subjects in Fig. 3. Perhaps plotting the average across 171 subjects could illustrate the goodness of the fit and highlight the performance of seq2seq model.

For the seq2seq model (Fig. 3), it would be illustrative to show loss functions to see if the model converged. If not, would a different learning rate or optimizer improve the convergence?

In Fig. 3 the amount of data that is being fitted as a part of test set is only 5%. The results could be more representative if the authors conduct iterative cross-validation analysis.

Based on previous work, authors claim "The architectural constraints of CMR determine that the only reliable way to recall items in a chaining manner, whether forward or backward, is to do so forward". It would be helpful to elaborate and justify this, at least on an intuitive level.

In Fig. 4. it would be helpful to see errorbars.

Fig. 4. shows different stages of training (each epoch) but does not show how the loss changed across training. In particular, since the authors used PPO optimization, they could look into both actor loss and critic loss.

It would be relevant to discuss whether seq2seq model would be capable to account for data in delayed free recall and free recall with continuous distractors.

Page 16 mentions supervised pretraining before RL, but does not provide details.

Regarding training the RL model, the authors say: "In order to train the seq2seq model independent of the order of recall, we treat the problem as a set prediction task and opt to use an iterative Sinkhorn algorithm approximation method that impacts training time, but ensures that network learning is not impacted by token order." Please provide more details and exact loss function and parameters that were used.

There is a number of recent paper that address serial position effects in sequence to sequence models (although typically in transformer models that do not use recurrent connections, but still capture many of the properties reported in free recall, including recency, primacy and free recall). It would be helpful to discuss similarity and relevance of the current work in the context of that previous work:

Li, Ji-An, et al. "Linking in-context learning in transformers to human episodic memory." *Advances in Neural Information Processing Systems* 37 (2025): 6180-6212.

Mistry, Deven Mahesh, et al. "Emergence of Episodic Memory in Transformers: Characterizing Changes in Temporal Structure of Attention Scores During Training." *arXiv preprint arXiv:2502.06902* (2025).

Guo, Xiaobo, and Soroush Vosoughi. "Serial position effects of large language models." *arXiv preprint arXiv:2406.15981* (2024).

Fountas, Zafeirios, et al. "Human-like episodic memory for infinite context llms." *arXiv preprint arXiv:2407.09450* (2024).

Overall, I believe this is quite solid work that can be useful to the community.

Reviewer #2 (Remarks to the Author):

Salvatore and Zhang develop a variant of a model conventionally applied to machine learning problems of prediction, the seq2seq model. They use this model to explain cognitive processes, rational explanations, and behavioral effects of human free recall. For most articles presenting a new model, at most they may compare the novel model's predictions to another model and at least they may just present the model anew. Yet here the authors provide intuition for the structural and conceptual overlap between the seq2seq model and an established cognitive memory model of free recall, the context maintenance and retrieval (CMR) model. In so doing, the authors bridge the seq2seq model to explain free recall effects in

healthy adults and amnesic patients, and provide novel intuitions of the cognitive underpinnings of episodic memory. These simulations pave the way for future developments to leverage the strengths of neural machine translation and cognitive models to characterize human memory search processes. Overall, I think this manuscript has promise to provide an innovative advancement of human memory models and processes. However, before recommending the manuscript for publication I think the authors need to address several key points. I list my concerns in descending order of importance.

1. The authors emphasize the importance of the attention mechanism in the seq2seq model in order to make accurate and optimal predictions of human behavior. However, this mechanism could be defined more clearly. The lefthand term in Equation 6 forms the key component of the attention mechanism in Equation 7, but this term is never formally defined and rather it seems to be recursive with attentional weights. This in turn provides little intuition for how or why the attention weights are assigned. Perhaps the development of the attention weights are adjusted based on the algorithm and may not be straightforward to define. In that case, might this mean that the model assumes that the human attentional system adjusts memory strengths either optimally or to match task performance? If so, how and why could the human memory system and attentional system interact to give rise to these weights? This would be in contrast to other cognitive memory models like CMR which provide a cognitive explanation the attentional weighting based on novelty, time shared in a short-term memory buffer, etc.

Thus, I think the authors need to do more work to clarify conceptually and mathematically how and why the attentional weights are developed to give rise to the current predictions. It might also help to say more about how the bias vector and activation function are calculated in Equation 2.

2. The authors describe that their final simulations were meant “to examine if models with and without attention could capture the key behavioral differences in the healthy controls and amnesic patients respectively.”

a. However, because the model simulations only indicate the optimal behavior not the actual patient behavior, I had more difficulty following how the model truly captures the differences, as opposed to just a rational explanation to predict idealized behavior. Notably, this optimal version of the model makes less accurate predictions of recall by serial position. Instead, I think the authors should include seq2seq predictions aligned with human behavior (i.e., Figure 6A-C), but perhaps more explanation of the strengths of the current approach could resolve the issue.

b. Although the authors “hypothesize that the lack of the attention mechanism (i.e. the ability to reinstate previous study contexts) makes it difficult to reinstate the beginning-of-list context, which is necessary to establish primacy”, the predicted level of primacy seems to be the same for the optimized model with or without attention, as shown in probability of first recall and the serial position curve in Figure 6. Recall probability also looks about the same for both groups with hidden dimension size of 128. I’m sorry if I’m missing something obvious here but I would appreciate more intuition on these discrepancies, especially given the centrality of attention to the manuscript.

3. I think the authors need to contextualize the seq2seq model (and CMR) in the broader modeling literature.

In the Discussion, it would be ideal to discuss how the seq2seq model relates to computational models of episodic memory aside from CMR, as well as other neural network models. To do this, I think the authors should be a bit more careful about distinguishing assumptions of CMR from the accepted ground truths of episodic memory. For instance, the authors assert that “Recalled items during a free recall task are driven by the current context, which is a combination of an evolving context from the previous time step (as maintained from the working memory) and a context reactivated from the encoding period (as retrieved from episodic memory)”. However, not all models of free recall assume that context plays such a central role or is reactivated during recall. Further, noting the strengths of other models could also help to qualify what other benchmarks might remain to make a stronger case that the seq2seq model outperforms CMR or other cognitive episodic memory models.

4. If possible I would prefer more explanation of the critical relationship in the seq2seq model between hidden dimension size and working memory. For instance, does 128 approach average human capacity, or more generally might these numbers translate to the number of studied items held in working memory?

5. The authors describe a weakness of CMR as “it can determine the combination of the two contexts in a single parameter but cannot derive the dynamic interaction of these two contexts and their relation to free recall performance”. The authors explain what I think they refer to as the dynamic interaction by allowing dimension size to vary in the seq2seq model. If that is the case, in the CMR simulations the authors only explore one parameter set under one set of assumptions. I wonder if CMR were given the chance to vary in order to explain these dynamics as well, then it might account for dynamics like the seq2seq model.

6. For any figures where values on the x and y axes overlap, perhaps the authors could offset the values on the x-axis to clarify this overlap. I wonder if this is why the “With Attention - Optimized” dark blue line appears minimally in Figures 6E,F.

7. The authors describe CMR as having a “latent contextual representation”, but isn’t the contextual representation explicitly defined (e.g., Equation 3)?

8. I got a bit lost as to why it is informative to characterize the development of optimal weights across epochs for the seq2seq model, as opposed to only the final results. Don’t the earlier epochs partially rely simply on how the model is initialized?

9. On page 10 when the authors refer to the “dashed line in Figure 4A-C”, I think they mean 4D-F.

Reviewer #3 (Remarks to the Author):

Salvatore & Zhang present integration of seq2seq models used in ML with theories of human memory search, specifically the Context Maintenance and Retrieval (CMR) model. Through reanalysis of previously published data, the authors show that the seq2seq model performs remarkably well, and with greater accuracy than the CMR model. Additionally, by relating seq2seq with the CMR model, the authors are able to provide interpretability, for instance, to show that a reduced WM capacity would predict greater reliance on episodic memory.

Overall, I really enjoyed the aim of the paper in relating descriptive psychological models to normative ML approaches. However, my enthusiasm was a bit dampened by issues due to clarity in both the figures and text in making these connections salient. Additionally, I had some concerns about the results, particularly in terms of the boundary conditions where seq2seq beats CMR. I outline these concerns below in more detail.

Clarity.

My main issue with clarity is due to the fact that the seq2seq and CMR use different notation, making it hard at times to relate similar mechanisms. I appreciate that figures 1 and 2 try to use similar colors to group related processes, but I think this could still be further improved. For instance, by making the meanings of the colors more explicit via figure legends, or to group sections of the model in Figure 2 into higher-level "modules", which could be enclosed in a box. This would allow for a shared schematic between the two approaches at the level of modules in order to provide a higher-level perspective.

But perhaps most importantly, it was hard to follow Eqs 1-9 due to jumping back and forth between the different notations used in each model. Some potential solutions could be to a) walk through each approach separately before integrating them together or b) develop a unified notation for both, which can be translated into the typical domain-specific notation in the SI. Either way, the current structure requires keeping in mind both notations simultaneously, which hurt my ability to appreciate some of the more subtle aspects of this theory integration and may reduce the accessibility of this work for non-specialized audiences.

Results.

One of the main findings is that seq2seq outperforms CMR in fitting human data. However, in which data regimes would this hold true? For instance, if you gave seq2seq less training data, would the performance cross-over at some point below CMR? Thus, it would be important to more carefully quantify the interpretation of the results to run analyses using different amounts of training data to see where the boundary conditions are.

Currently, the results make this difficult to assess. Details are missing about how cross-validation is performed. The statistics on page 8 lack DOFs. And why does Figure 3 present these specific 12 subjects?

Additionally, Figure 4D is never explained in the text, even though I found this to be one of the most interesting results. There may also be some other mistakes in figure references, since the text jumps from referencing Figure 4c to Figure 4G, omitting the entire 2nd row. On page 10 there is a reference to dashed lines in Figure 4A-C, but there are no lines.

If you experience problems in linking your ORCID, please contact the Platform Support Helpdesk.

Version 1:

Decision Letter:

Dear Dr. Zhang,

Your manuscript titled "Sequence-to-Sequence Models with Attention Mechanistically Map to the Architecture of Human Memory Search" has now been seen by our reviewers, whose comments appear below. In light of their advice I am delighted to say that we are happy, in principle, to publish a suitably revised version in Communications Psychology.

We therefore invite you to revise your paper one last time to address the remaining concerns of our reviewers and a list of editorial requests. At the same time we ask that you edit your manuscript to comply with our format requirements and to maximise the accessibility and therefore the impact of your work.

EDITORIAL REQUESTS:

SUBMISSION INFORMATION:

OPEN ACCESS:

*** TRANSPARENT PEER REVIEW:** Communications Psychology uses a transparent peer review system. On author request, confidential information and data can be removed from the published reviewer reports and rebuttal letters prior to publication. If you are concerned about the release of confidential data, please let us know specifically what information you would like to have removed. Please note that we cannot incorporate redactions for any other reasons.

*** CODE AVAILABILITY:** All Communications Psychology manuscripts must include a section titled "Code Availability" at the end of the methods section. We require that the custom analysis code supporting your conclusions is made available in a publicly accessible repository at this stage; please choose a repository that generates a digital object identifier (DOI) for the code; the link to the repository and the DOI must be included in the Code Availability statement. Publication as Supplementary Information will not suffice.

*** DATA AVAILABILITY:**

Link Redacted

Best regards,

Troy Lui

Troy Lui, PhD
Associate Editor
Communications Psychology

REVIEWERS' COMMENTS:

Reviewer #1 (Remarks to the Author):

The authors did a thorough job addressing my suggestions and concerns. I understand that performing iterative cross-validation on the entire dataset would be computationally too demanding. My only minor remark concerns Figure 3F, where the forward asymmetry, a key feature in the human data, appears to be nearly gone in Seq2Seq.

Reviewer #2 (Remarks to the Author):

I am grateful to the authors for addressing my concerns and clarifying my points of confusion. After reviewing these points as well as their other revisions, I recommend the manuscript for publication.

Reviewer #3 (Remarks to the Author):

My concerns relating to clarity have been greatly improved in the revised version of the manuscript, particularly with the new Fig. 2. The updated equations certainly help as well and are quite thorough, although the authors may decide on their own whether to move some elements to the methods or SI to prevent hurting the flow of the paper.

I'm also very happy with how the authors addressed my concern about the generalizability of the results in terms of the amount of training data. One very minor point about Figure S3 is that in panels B-D it's difficult to tell which model wins. Fig. S2 is also quite difficult to assess, and it might be better to plot some aggregate means (e.g., boxplots by split) rather than the current bar plots separated by subject.

All my other concerns have also been addressed in the revision.

Reviewer #4 (Remarks to the Author):

The authors of this manuscript compare the similarities in architecture of a seq2seq model with an attention mechanism and a CMR model. Their ultimate goal is to use this novel seq2seq model in human cognition, namely memory. They showcase the similarities in task performance between the two model types in a list learning task compared to human performance, quantify the relationship between attention/primacy effects and working memory capacity, and leverage the attention mechanism in the seq2seq model to possibly explain task performance for amnesia patients. Together, these results show that a seq2seq model with an attention mechanism is similar to a CMR model's context mechanism, suggesting that this type of seq2seq model can be used to accurately model human memory in both healthy controls and those with MTL-based amnesias.

The way that the authors break down the similarities in the architectures of their proposed RNN and a CMR model was very easy to follow. The list-learning task clearly showed how their RNN model with attention performs similarly to a CMR model. One of the most interesting uses of the model was the simulation of memory deficits in amnesia, which shows that this model is useful not for just modeling healthy controls but also for patient groups. In reviewing the comments to the previous submission of this manuscript, the authors have been very thorough in their responses, which have made their work extremely clear. I have no additional comments, and this manuscript will be a great addition to the literature on the use of computational models to explain human memory.

The code is very well commented out with an easy to follow README document. It functions well and is very detailed, which helps to easily reproduce their results.

— Reviewer 1 —

R1.1 - Eq. 4 might contain an error - it states that both item to context and context to item matrices are identical. Shouldn't item to context association be $f_i c_{i-1}^T$ and context to item $c_{i-1} f_i^T$. More explicit mention of dimensions for each of the vectors and matrices would also help with understanding.

Thank you for pointing this out. We have amended the description of the item-context association matrices to include two separate equations correctly denoting the definitions. The amended sections are as follows:

P7, L197:

$$\Delta M_{\text{exp}}^{FC} = c_{i-1} f_i^T$$

Similarly, the association from context to item, M_{exp}^{CF} , is updated according to:

$$\Delta M_{\text{exp}}^{CF} = f_i c_{i-1}^T$$

Additionally, we have added explicit mentions of the dimensions of the vectors and matrices discussed.

P4, L144:

“Each input item f_i presented at timestep i is embedded into a dense vector $x_i \in \mathbb{R}^d$ ”

“the encoder RNN processes the sequence one item at a time, updating its hidden state $h_i \in \mathbb{R}^d$ ”

“the model also stores the concatenation of all hidden states, $H = [h_1, h_2, h_3, \dots, h_L]$ ”

“the decoder RNN receives the embedding $x_j \in \mathbb{R}^d$ ”

“along with the previous hidden state $h_{j-1} \in \mathbb{R}^d$ ”

“ $\alpha_j = \sum_{i=1}^L w_j^i h_i \in \mathbb{R}^d$ ”

“Here, $W_c \in \mathbb{R}^{d \times 2d}$ are the learned parameters of the mixing layer”

“ $f_j = \psi(\hat{h}_j) \in \mathbb{R}^N$ ”

“participants study a list of L items one after another (drawn from a total number of N possible items in the experimental word pool)”

“The state of the context at time step $c_i \in \mathbb{R}^N$ ”

“where $x_i \in \mathbb{R}^N$ is the retrieved context (or input embeddings)”

“ $M_{\text{pre}}^{FC} \in \mathbb{R}^{N \times N}$ ”

“ $f_i \in \mathbb{R}^N$ ”

“ $M_{\text{exp}}^{FC} \in \mathbb{R}^{N \times N}$ and $M_{\text{exp}}^{CF} \in \mathbb{R}^{N \times N}$ ”

“ $a_j \in \mathbb{R}^N$ ”

R1.2 - Please provide more information about the neural network and the training process: number of parameters, learning rate, optimizer.

We have added more detailed explanations of the neural network and the exact training procedure to the Methods section found near the end of the paper.

The new paragraph added is as follows:

P18, L550: “For all seq2seq models in both the individual fitting and optimization experiments, the encoder and decoder GRUs each use a hidden dimension size of 32, 64, or 128, depending on the experiment, and the word embeddings are 50-dimensional pre-trained GloVe vectors. The total number of trainable parameters in the seq2seq model across experiments is 99,184, 162,704, and 332,752 for the 32, 64, and 128 hidden dimension-sized models, respectively.”

P18, L576: “For this supervised training, each model is trained using the Adam optimizer with a learning rate of 0.001, $\beta_1 = 0.9$, and $\beta_2 = 0.999$. Training was performed using mini-batches of size 32 with an early stopping patience of 5 epochs, allowing each model to train for an unspecified number of epochs required until validation loss failed to improve for 5 epochs.

No dropout or gradient clipping was used in training across any of the experiments.”

R1.3 - Is there anything specific in the attention computations - would you expect different results if the attention layer was replaced with dense?

Thank you for the suggestion. We have trained and evaluated a seq2seq model where the attention layer was replaced by a dense layer on the decoder hidden state (hidden dim = 64), with no dot product or access to encoder states. We have plotted recall behavior for each of the recall metrics alongside the models from our original analysis and included these new results in the supplementary material.

Our additions are as follows:

SP1, L3: “**Examining Alternative Model Architectures without Attention** - To evaluate whether the benefits of the attention module arise from its underlying mechanism or merely from its increased parameterization or model complexity, we trained and evaluated a seq2seq model in which the standard attention layer was replaced by a dense layer operating only on the decoder hidden states (with no access to encoder states). We then repeated the same analysis reported in the main results in Figure 6. As shown in Figure S1 A—C, the dense-layer model produced serial position, probability of first recall, and conditional response probability that were qualitatively similar to those of the model without attention (i.e., with the standard attention layer removed). Both models exhibited deficits in recall performance (Figure S1 A) and showed no backward contiguity effects in any stage of training (Figure S1 C) as compared to the model with the attention mechanism. These results confirm that the advantage of attention in our framework is its ability to access prior encoding contexts, not simply due to increased parameterization or model complexity.”

Figure S1: Behavioral patterns of the seq2seq model with attention, without attention, and with the attention module replaced by a dense layer operating only on the decoder hidden states, optimized or during intermediate training.

Models are compared across three sets of patterns: serial position curve (A), probability of first recall (B), and conditional response probability (C). The seq2seq model with a dense layer shows similar performance and behavior to the model without the attention mechanism. Both models show reduced recall performance (A) and a lack of backward contiguity as compared to the model with attention (C). Model hidden dimension size is 64.

R1.4 - The statistical analysis on page 8 leads to a conclusion that “seq2seq model fits the corresponding human data curve with significantly smaller error for serial position, probability of first recall and conditional response probability”. This is not particularly apparent from the data from 12 selected subjects in Fig. 3. Perhaps plotting the average across 171 subjects could illustrate the goodness of the fit and highlight the performance of seq2seq model.

Thank you for this suggestion. We have now added plots averaged across 171 subjects to the Results section 1. We have acknowledged that both for individual subjects (first 12 plotted) or averaged across subjects, CMR and the seq2seq model have similarly good qualitative fits to the human recall data. However, if we quantitatively evaluate their fits in terms of the root-mean square error between the human data and model simulations across all 171 subjects, we can see from the Figure 3D-F that the seq2seq model has smaller root-mean square errors across all three set of behavioral patterns. Our goal here is not to show that the seq2seq model can capture certain qualitative patterns that CMR cannot, but to support that “The seq2seq model with attention can capture human recall patterns as effectively as CMR”.

Figure 3. Behavioral patterns for individual subjects and the model predictions. (A–C) Behavioral patterns for the first 12 subjects out of 171 subjects, reproduced from the Experiment 1 of PEERS free recall dataset⁴⁹, overlaid with behavioral patterns of the CMR model and the seq2seq model with attention, trained over a separate subset of the same individual subject data. The behavioral patterns include (A) the serial position curve, (B) the probability of the first recall, and (C) the conditional response probability. (D–F) Aggregated behavioral patterns for all 171 subjects overlaid with aggregated CMR and seq2seq model behavioral patterns. Model fits are quantitatively evaluated regarding the root-mean-square error between the human behavioral patterns of the 171 subjects and model predictions over (G) the serial position curve, (H) the probability of the first recall, and (I) the conditional response probability.

These clarifications and new results on the averaged data have been added to the Results section:

P10, L289: “Figures 3A–C depict the behavior patterns of the first 12 subjects (for illustrative purposes) of the 171 subjects taken from the PEERS dataset with both fitted CMR and seq2seq models overlaid. Additionally, we show the average recall behavioral patterns for all 171 subjects in Figures 3D–F. While both CMR and the seq2seq model have qualitatively good fits to the human recall data, we compare their model fit quality in terms of the root-mean square error between the human recall behavior and corresponding model recall behavior across all 171 subjects (Figure 3G–I). [...] Together, these results support that the seq2seq model with attention can capture human recall patterns as effectively as CMR. This ability to capture human recall patterns is not a result of the neural network model being over-parameterized as the trained model was able to predict behavioral patterns in the unseen portion of recall data.”

R1.5 - For the seq2seq model (Fig. 3), it would be illustrative to show loss functions to see if the model converged. If not, would a different learning rate or optimizer improve the convergence?

Thank you for suggesting this analysis. We have checked the convergence of models for all subjects and plotted below the loss function for the first 20 subjects to illustrate the convergence.

Training and Validation Losses for Individual Subject Fitting

Training and Validation Loss Curves for Individual Subject Fitting. Training and validation loss curves for individual subject models indicate convergence of the seq2seq models with attention across the first 20 subjects illustrated.

We have added a summary of this convergence result and details about the learning rate and optimizer to the Methods section in the main text:

P18, L579: “For this supervised training, each model is trained using the Adam optimizer with a learning rate of 0.001, $\beta_1 = 0.9$, and $\beta_2 = 0.999$. Training was performed using mini-batches of size 32 with an early stopping patience of 5 epochs, allowing each model to train for an unspecified number of epochs required until validation loss failed to improve for 5 epochs. No dropout or gradient clipping was used in training across any of the experiments. Training and validation loss curves indicated convergence of individual subject models.”

R1.6 - In Fig. 3 the amount of data that is being fitted as a part of test set is only 5%. The results could be more representative if the authors conduct iterative cross-validation analysis.

Thank you for raising this important issue. We agree that cross-validation can provide a more comprehensive assessment of the seq2seq model’s improved ability to fit individual human recall behavior . As a result, we’ve run additional cross-validation analyses to show that the same conclusion holds while using cross-validation compared with the 90-5-5 split.

Performing this analysis on the entire dataset was impractical given time and compute constraints. A full 5-fold cross-validation run for a single subject requires on average 17.5 hours of training time. Extending this procedure to all 171 subjects in our dataset would require over 3,000 GPU-hours (i.e. almost 125 days), making a full cross-validation analysis of the entire dataset infeasible with our resources and time limitations. To address this, we have randomly chosen 20 subjects from the origi-

nal dataset on which to perform the 5-fold cross-validation study. The results added to the supplementary material are as follows:

SPI, L13: “Cross-validation Study of Individual Subject Fitting - To assess whether different methods of conducting the training and testing splits would affect our conclusion that seq2seq performs better than CMR in predicting recall behavior, we conducted a 5-fold cross-validation on a random subset of 20 subjects from the full dataset of 171 subjects given our time and compute constraints (A full 5-fold cross-validation run for a single subject requires on average 17.5 GPU-hours). In this cross-validation, the dataset is randomly divided into five equal parts, and the model is trained on four parts while the remaining part is used for validation and testing; this process is repeated five times so that each part serves as the validation and test set once. To maintain consistency with our original 90%-5%-5% train-validation-test splits, we used the standard 5-fold cross-validation with an 80%-20% train-test split. After designating 80% of the data for training in each fold, we randomly sampled two non-overlapping 5% subsets – one for validation and one for testing – from the remaining 20% of data. The seq2seq model with attention was trained and evaluated using the same procedures as in the main analysis. Model performance was assessed on held-out folds using root-mean-square error (RMSE) across the three behavioral measures (the serial position curve, the probability of first recall, and the lag-conditional response probability) as in our original analysis depicted in Figure 3. Figure S2 shows the RMSE for each cross-validation fold relative to the corresponding subject’s recall patterns (averaged across all behavioral measures), with the RMSE for CMR included for comparison.

We see that a significant difference between seq2seq and CMR RMSE remains after the cross-validation study, corroborating the results of our main analysis. To ensure this difference was significant, we conducted a Wilcoxon signed-rank test comparing the average of the RMSE means (i.e., the average RMSE for each of the three behavior measures, averaged across all seq2seq model splits) with the CMR RMSEs (two-sided: $W = 0, n = 20, p = 1.907 \times 10^{-6}$). This result indicates that the seq2seq models exhibit significantly lower RMSE values on average compared to CMR. These findings support that different methods of conducting the training and testing splits do not affect our conclusions in model comparison.

Figure S2: 5-Fold cross-Validation results of individual subject fitting for the seq2seq model with attention and CMR. Model performance was assessed on held-out folds using root-mean-square error (RMSE) across all three behavioral measures (the serial position curve, the probability of first recall, and the lag-conditional response probability).

A mention in the main text has been added as follows:

P10, L299: “These results are not affected by different methods of conducting the training and testing splits (see Supplementary Materials S2).

R1.7 - Based on previous work, authors claim “The architectural constraints of CMR determine that the only reliable way to recall items in a chaining manner, whether forward or backward, is to do so forward”. It would be helpful to

elaborate and justify this, at least on an intuitive level.

Thank you for pointing this out. We have added more context in explaining the intuition behind this result:

P10, L315: “ Why is it necessary to start recalling from the beginning and recall forward, rather than from the end and then recall backward? In CMR, each item is encoded along a gradually drifting internal context. During recall, retrieving an item brings back the context from when that item was originally studied, which then serves to cue recall of other items. During this process, there is a reliable way to proceed forward (by reinstating the pre-experimental context associated with the item), but there is not a reliable way to proceed backward — the only way to promote backward transitions is to reinstate the experimental context associated with the item at study, which is equally likely to propel recall forward and backward. As a result, it yields better performance when initiating the recall from the beginning of the list and then recalling forwards³¹.”

R1.8 - In Fig. 4. it would be helpful to see error bars.

Thank you for the suggestion. We have added error bars to all data series in Fig. 4 with the exception of the rational CMR curve, which was reproduced from previous work representing one single simulation. Error bars for the Human Subject data indicate inter-subject variability, whereas the error bars for the seq2seq model indicate inter-trial variability. The error bars are extremely small for the seq2seq model. We have updated Figure 4 in the manuscript to reflect these changes:

Figure 4. Optimized and intermediate training of the Seq2Seq model with attention. The fully optimized seq2seq model with attention (128-Dim) shows behavioral patterns that closely align with those of the rational-CMR model across three sets of free recall patterns: serial position curve (A), probability of first recall (B), and conditional response probability (C). Rational-CMR results are reproduced from Zhang 2023 et. al. ³¹. (D-F) Recall characteristics are evaluated for intermediate epochs of training and compared qualitatively to the average subject recall from the PEERS dataset. Intermediate model results exhibit recency effects and backward contiguity effects that are typical in human behavioral patterns. (G-I) The same results are plotted along the dimension of training epochs, summarizing the model’s tendency to initiate recall from the end (the last three items of the list; G), to initiate recall from the beginning of the list (the first three items of the list; H), and to recall items in the backward direction (conditional response probability with -1 lag; I). (J–L) Actor and critic loss curves exhibit a consistent decrease over time, and the mean episode reward converges toward the maximum possible recall reward of 14 for our experiments. 5,000 training steps correspond to 1 training epoch.

R1.9 - Fig. 4. shows different stages of training (each epoch) but does not show how the loss changed across training. In particular, since the authors used PPO optimization, they could look into both actor loss and critic loss.

Thank you for this suggestion. In order to provide a full picture of the model’s optimization process, we have updated Figure 4 and included training loss curves and mean episode reward curves to our main results (Figure 4 J-L).

The updated Figure 4 is pictured below with mentions of the loss curves:

P10, L323: “[...]reproduced from Zhang et al. ³¹). We additionally plot the actor and critic losses along with the mean episode reward across training (Figure 4J–L) to demonstrate convergence of the seq2seq model with attention. Both models achieve near-optimal performance [...]”

Figure 4. Optimized and intermediate training of the Seq2Seq model with attention. The fully optimized seq2seq model with attention (128-Dim) shows behavioral patterns that closely align with those of the rational-CMR model across three sets of free recall patterns: serial position curve (A), probability of first recall (B), and conditional response probability (C). Rational-CMR results are reproduced from Zhang 2023 et. al. ³¹. (D-F) Recall characteristics are evaluated for intermediate epochs of training and compared qualitatively to the average subject recall from the PEERS dataset. Intermediate model results exhibit recency effects and backward contiguity effects that are typical in human behavioral patterns. (G-I) The same results are plotted along the dimension of training epochs, summarizing the model’s tendency to initiate recall from the end (the last three items of the list; G), to initiate recall from the beginning of the list (the first three items of the list; H), and to recall items in the backward direction (conditional response probability with -1 lag; I). (J-L) Actor and critic loss curves exhibit a consistent decrease over time, and the mean episode reward converges toward the maximum possible recall reward of 14 for our experiments. 5,000 training steps correspond to 1 training epoch.

R1.10 - It would be relevant to discuss whether seq2seq model would be capable to account for data in delayed free recall and free recall with continuous distractors.

Thank you for suggesting this idea. We agree that extending the seq2seq model to account for delayed free recall and free recall with continuous distractors would be an interesting and important direction for future work.

We have acknowledged this limitation of our current analysis and added discussions under the section “An alternative model of human memory search”:

P16, L459: “Our analysis reveals that the seq2seq model with attention serves as an effective substitute for the CMR model in fitting and predicting individual free recall characteristics. The seq2seq model not only aligns closely with human recall patterns but also outperforms CMR in certain predictive tasks. This supports the idea that the seq2seq approach could be

adopted as a useful tool in cognitive modeling for understanding human memory search. [...]

One limitation of our current study is that we evaluate our model’s ability to predict recall patterns over one type of free recall task when participants are asked to recall immediately after the encoding phase. There are other variants of the free recall task, where participants are required to recall the list after a delay period⁶⁶ – introducing a context change between encoding and recall, or where there are distractor tasks in between every pair of adjacent words during encoding⁶⁷ – introducing a context change between encoding adjacent words. CMR can capture these tasks through the amount of context drift. We expect that the seq2seq model can similarly account for these tasks, given the mapping we have derived between the seq2seq model architecture and the context drifting mechanism in CMR, and the flexibility in the seq2seq model to learn from the complex dynamics of context change. It remains a fruitful venue in the future to evaluate the seq2seq model to account for memory phenomena in different variants of the free recall task. Another limitation in the current work is our focus on examining the relationship between the seq2seq model with one specific memory search model, CMR. In addition to the CMR model, there is another prominent model of memory search, the Search of Associative Memory (SAM) model⁶⁸, which relies on the distinction between a short-term store and a long-term store without reactivating prior contexts at retrieval. Both SAM and CMR have a history of successfully accounting for various behavioral patterns in memory search. Despite similar levels of empirical evidence they receive, our present work in aligning CMR with the seq2seq model in the machine learning community provides additional support to the CMR model in terms of its computational efficiency.

R1.11 - Page 16 mentions supervised pretraining before RL, but does not provide details.

Thank you for pointing this out. Our response is combined with the next comment.

R1.12 - Regarding training the RL model, the authors say: “In order to train the seq2seq model independent of the order of recall, we treat the problem as a set prediction task and opt to use an iterative Sinkhorn algorithm approximation method that impacts training time, but ensures that network learning is not impacted by token order.” Please provide more details and exact loss function and parameters that were used.

Full details of the supervised pre-training and set prediction loss function have been included in the expanded Methods section. The additions are shown below:

P19, L592: **“Supervised Pre-Training:** We pre-trained the seq2seq models in a supervised fashion using randomly generated lists of words appearing in the PEERS dataset vocabulary. This supervised pre-training phase initializes the seq2seq models with useful representations and stable behaviors before reinforcement learning, improving sample efficiency and mitigating exposure bias—where models perform significantly better on input items that appeared more frequently in training—which has been shown effective in sequence prediction and neural machine translation tasks^{81,82}. The model was trained to output the entire recall set for each input list, irrespective of item order. To accomplish this, we used a set prediction loss based on the differentiable Sinkhorn algorithm^{83,84}. For each training batch, we first computed the cross-entropy loss between each predicted token and every target token, resulting in a cost matrix C . The Sinkhorn-Knopp algorithm was then applied to $-C/\tau$, where the temperature parameter τ (set to 1.0 in our experiment) controls the sharpness of the soft assignment between predictions and targets. After 20 Sinkhorn iterations, this yields a doubly-stochastic matching matrix P , and the final set loss is defined as: $L_{\text{set}} = \frac{1}{B} \sum_{b=1}^B \sum_{i=1}^N \sum_{j=1}^N P_{ij}^{(b)} C_{ij}^{(b)}$, where B is the batch size and N is the sequence length. We used a batch size of 32, Adam optimizer with a learning rate of 0.001, and trained for a total of 10 epochs. No dropout or gradient clipping was used in training across any of the experiments. For comparison of optimal model behavior and human data, we use the vocabulary from experiments in the PEERS free recall dataset⁴⁹. For pre-training, all model configurations were trained using 50,000 randomly generated sequences of 14 words (no repeats) sampled from the PEERS vocabulary. The weights learned during this pre-training phase were used to initialize the model for subsequent reinforcement learning.”

R1.13 - There is a number of recent paper that address serial position effects in sequence to sequence models (although typically in transformer models that do not use recurrent connections, but still capture many of the properties reported in free recall, including recency, primacy and free recall). It would be helpful to discuss similarity and relevance of the current work in the context of that previous work:

Li, Ji-An, et al. “Linking in-context learning in transformers to human episodic memory.” *Advances in Neural Information Processing Systems* 37 (2025): 6180-6212.

Mistry, Deven Mahesh, et al. “Emergence of Episodic Memory in Transformers: Characterizing Changes in Temporal Structure of Attention Scores During Training.” arXiv preprint arXiv:2502.06902 (2025).

Guo, Xiaobo, and Soroush Vosoughi. “Serial position effects of large language models.” arXiv preprint arXiv:2406.15981 (2024).

Fountas, Zafeirios, et al. “Human-like episodic memory for infinite context llms.” arXiv preprint arXiv:2407.09450 (2024).

We agree that these recent papers have particular relevance to our work and have included an additional Discussion section that highlights their findings, particularly with respect to how transformer-based models capture primacy, recency, and contiguity effects.

P16, L462: “While the current work focuses on RNN-based models, Large Language Models (LLMs) offer another promising avenue for modeling human memory. There is evidence that pre-trained LLMs exhibit striking similarities to human episodic recall, including serial position effects^{61,62} and temporal contiguity^{63,64}, without being explicitly trained on memory tasks. Recent work suggests some of these behaviors, particularly the temporal contiguity effects, emerge from the activity of induction heads within the transformers, which have been directly mapped to mechanisms in cognitive models of human memory like the Context Maintenance and Reinstatement (CMR) model⁶³. Additionally, ablating these specific heads eliminates the structured recall patterns⁶⁴. Identifying convergences between human memory and machine learning models can not only deepen understanding of human memory but also drive improvements in machine learning algorithms. In one such study, Fountas et al.⁶⁵ improve the LLM’s ability to effectively use its context by explicitly integrating key aspects of human episodic memory and event cognition into the model architecture.”

— Reviewer 2 —

R2.1 - The authors emphasize the importance of the attention mechanism in the seq2seq model in order to make accurate and optimal predictions of human behavior. However, this mechanism could be defined more clearly. The lefthand term in Equation 6 forms the key component of the attention mechanism in Equation 7, but this term is never formally defined and rather it seems to be recursive with attentional weights. This in turn provides little intuition for how or why the attention weights are assigned. Perhaps the development of the attention weights are adjusted based on the algorithm and may not be straightforward to define. In that case, might this mean that the model assumes that the human attentional system adjusts memory strengths either optimally or to match task performance? If so, how and why could the human memory system and attentional system interact to give rise to these weights? This would be in contrast to other cognitive memory models like CMR which provide a cognitive explanation the attentional weighting based on novelty, time shared in a short-term memory buffer, etc. Thus, I think the authors need to do more work to clarify conceptually and mathematically how and why the attentional weights are developed to give rise to the current predictions. It might also help to say more about how the bias vector and activation function are calculated in Equation 2.

Thank you for the thoughtful feedback. We’ve now clarified the definitions and roles of terms in the equations, particularly how the attention weights are computed using dot-product attention. These weights are calculated at each decoding step by comparing the current decoder hidden state to each encoder hidden state; the dot product serves as a similarity metric and is not recursively defined with any other aspect of the mechanism. Importantly, the attention weights are not fixed or rule-based; and they are learned during training to either match human recall patterns or to achieve optimal memory performance. Full details on the algorithms of how model parameters (including attention weights) are learned can be found under Methods in sections “Explain and Predict Individual Subject Behavior” and “Optimal Free Recall Behavior”. While the mechanism is referred to as “attention” by the machine learning community, it is not the same as the attention weights during human memory encoding, which change according to novelty or time shared in a short-term memory buffer. Rather, the mechanism takes effect at the recall phase and serves as a functional analogue to context reinstatement mechanism in models like CMR, allowing the model to selectively reinstate past encoding contexts based on their relevance to the current retrieval context.

We have modified the relevant sections to more explicitly define the variables and their relationships as well as presenting the inner workings of the model in a clearer manner:

P6, L159: During decoding, the attention mechanism (shown in the green box in Figure 2B) gives the decoder access to all encoder hidden states H via attention weights. These attention weights are obtained by calculating a score function between the

current decoder hidden state h_j and each hidden state h_i from the encoding stage. We use the dot product as the score function as specified in Luong attention⁸. to avoid introducing additional parameters. After calculating scores for each encoder hidden state, the softmax operation is applied to obtain the attention weights w as shown below:

$$w_j^i = \frac{\exp(h_j^\top h_i)}{\sum_{i=1}^L \exp(h_j^\top h_i)} \quad (1)$$

where $h_j^\top h_i$ is the dot product, i.e., the similarity between the current decoder hidden state j and an encoder hidden state h_i . The attention weight w_j^i refers to how much attention the model should pay to each encoder hidden state h_i relative to all other encoding states at the decoding step j , which represents the importance of the encoder hidden state h_i to producing the output at the decoding step j .

Once the attention weights have been computed, the overall attention context vector (α_j) is obtained via a weighted sum of encoder states:

$$\alpha_j = \sum_{i=1}^L w_j^i h_i \in \mathbb{R}^d \quad (2)$$

Before an output is generated, a final hidden state vector \hat{h}_j is formed by combining the decoder hidden state and the attention context through a dense, mixing layer:

$$\hat{h}_j = \tanh(W_c [h_j; \alpha_j]) \quad (3)$$

Here, $W_c \in \mathbb{R}^{d \times 2d}$ are the learned parameters of the mixing layer, while $[h_j; \alpha_j]$ is the concatenation of the current decoder hidden state h_j and the attention context vector α_j .

An output, f_j , is generated at the decoding step j based on an output/retrieval rule ψ in conjunction with the final hidden state \hat{h}_j :

$$f_j = \psi(\hat{h}_j) \in \mathbb{R}^N \quad (4)$$

where $f_j \in \mathbb{R}^N$ is a one-hot column vector that is all zeros except at the position representing the item's identity, and N is the total number of possible items in the experiment. ”

R2.2 - The authors describe that their final simulations were meant “to examine if models with and without attention could capture the key behavioral differences in the healthy controls and amnesic patients respectively.” However, because the model simulations only indicate the optimal behavior not the actual patient behavior, I had more difficulty following how the model truly captures the differences, as opposed to just a rational explanation to predict idealized behavior. Notably, this optimal version of the model makes less accurate predictions of recall by serial position. Instead, I think the authors should include seq2seq predictions aligned with human behavior (i.e., Figure 6A-C), but perhaps more explanation of the strengths of the current approach could resolve the issue.

Thank you for raising this important issue. The intention of the ablation study was not to directly fit the recall behavior of MTL amnesia patients, but to understand what gives rise to performance difference between healthy controls and amnesia patients, which is why a rational approach is required. To clarify this point, the title of the section now reads: “An ablation study of the attention mechanism provides insight into the performance difference between amnesia patients and healthy controls.”

We hypothesize in the current work that MTL damage can be modeled by ablation of the attention mechanism in our model, and comparing an optimized model with or without attention can explain performance differences in healthy controls and amnesia patients. A rational approach not only reveals idealized behavior but also reveals the functional role of different cognitive processes in a model, because we can observe how they interact and contribute to final performance. If we just fit a model directly to two groups of human data, it can describe the behavior but does not elucidate why these differences occur and how they contribute to performance. For example, in the traditional modeling fitting approach with cognitive modeling (without optimization), different performance levels can often be trivially captured by increasing overall encoding strengths or removing

overall retrieval noises if we allow the full degree of freedom in fitting model parameters. In contrast, our model with or without attention has everything being identical except for the attention module removed or not– and behavioral differences in both recall performance and backward contiguity arise during the learning process without these two models directly fit to these patterns. We have added clarifications of our approach to the Results section to make the goal of the experiment more explicit:

P14, L389: “[...] As we hypothesized that the attention mechanism in our seq2seq model is analogous to context reinstatement in CMR, we carried out an ablation study of the attention module to examine if models with and without attention, when trained to optimize recall performance, could capture the key behavioral differences in the healthy controls and amnesic patients respectively. Results of this ablation are displayed in Figures 6D–F, alongside the model with attention, where both models are trained with a hidden dimension size of 64. We found similar results with the human data in the recall performance, where the attention model can recall more information than the no-attention model after being fully optimized (Figure 6D). Moreover, the lack of the attention mechanism in Figure 6F largely eliminates the model’s capacity for backward contiguity (at any stage of training) compared to the attention model, where backward contiguity is observed in early iterations of training. These ablation results support the idea that the attention mechanism in the seq2seq model is analogous to the context reinstatement mechanism in CMR (which has previously been linked with MTL damage³⁶) and contribute to memory performance during free recall. Importantly, the differences in recall performance and backward contiguity between attention and no-attention models do not come from direct fitting to human data but emerge during the process of optimizing performance, revealing the functional role of the attention mechanism in contributing to memory performance.”

R2.3 - Although the authors “hypothesize that the lack of the attention mechanism (i.e. the ability to reinstate previous study contexts) makes it difficult to reinstate the beginning-of-list context, which is necessary to establish primacy”, the predicted level of primacy seems to be the same for the optimized model with or without attention, as shown in probability of first recall and the serial position curve in Figure 6. Recall probability also looks about the same for both groups with hidden dimension size of 128. I’m sorry if I’m missing something obvious here but I would appreciate more intuition on these discrepancies, especially given the centrality of attention to the manuscript.

Thank you for pointing this out. You are correct that there is little clear difference in Figures 6D–F. This is due to the hidden dimension being large (128), allowing both models with and without attention to maintain sufficient working memory and recover primacy, as also shown in Figures 6G–H. To better illustrate the impact of attention on primacy, we have updated 6D–F to show results for the 64-dimensional model (instead of 128-dimensional model), where working memory capacity is more limited and the deficit from removing attention becomes evident:

Figure 6. Comparing optimized model behavior under ablation of the attention mechanism with recall behavior of medial temporal lobe (MTL) amnesia patients. (A–C) Behavioral patterns of MTL amnesia patients demonstrating memory deficits on a free recall task, reproduced from Palombo, et al³⁶. Despite the small list size, MTL amnesia patients exhibit diminished recall performance and reduced ability to perform backward recall transitions (conditional response probability at -1 lag). (D–F) Behavioral patterns of the seq2seq model with and without attention (hidden dimension size 64), optimized or during intermediate training. The model without attention exhibits diminished recall performance and reduced ability to perform backward recall transitions (no backward contiguity at any stage of training as seen from the conditional response probability at -1 lag). (G–H) Behavioral patterns of models of varying hidden dimension sizes with and without attention. Models without attention can achieve the optimal recall behavior but require sufficiently large hidden dimensions.

We have also updated the main text:

P14, L392: “Results of this ablation are displayed in Figures 6D–F, alongside the model with attention, where both models are trained with a hidden dimension size of 64.”

R2.4 - In the Discussion, it would be ideal to discuss how the seq2seq model relates to computational models of episodic memory aside from CMR, as well as other neural network models. To do this, I think the authors should be a bit more careful about distinguishing assumptions of CMR from the accepted ground truths of episodic memory. For instance, the authors assert that “Recalled items during a free recall task are driven by the current context, which is a combination of an evolving context from the previous time step (as maintained from the working memory) and a context reactivated from the encoding period (as retrieved from episodic memory”. However, not all models of free recall assume that context plays such a central role or is reactivated during recall. Further, noting the strengths of other models could also help to qualify what other benchmarks might remain to make a stronger case that the seq2seq model outperforms CMR or other cognitive episodic memory models.

Thank you for pointing this out. We have corrected our language so that the sentence only refers to CMR model rather than how memory works in general:

P17, L488: “Recalled items during a free recall task, as specified by the CMR model, are driven by the current context, which is a combination of an evolving context from the previous time step (as maintained from the working memory) and a context reactivated from the encoding period (as retrieved from episodic memory).”

We also acknowledged in the Discussions the limitation of the current study in focusing on one particular model of memory search CMR, while there is also SAM, which is another prominent model of memory search. We did not discuss exhaustively all the episodic memory models out there, as the focus of the paper is on free recall and memory search (which we also made it clear in our updated paper title “Sequence-to-sequence models with attention map to the architecture of human memory search”).

P16, L480: “Another limitation in the current work is our focus on examining the relationship between the seq2seq model with one specific memory search model, CMR. In addition to the CMR model, there is another prominent model of memory search, the Search of Associative Memory (SAM) model⁶⁸, which relies on the distinction between a short-term store and a long-term store without reactivating prior contexts at retrieval. Both SAM and CMR have a history of successfully accounting for various behavioral patterns in memory search. Despite similar levels of empirical evidence they receive, our present work in aligning CMR with the seq2seq model in the machine learning community provides additional support to the CMR model in terms of its computational efficiency and adaptability.”

R2.5 - If possible I would prefer more explanation of the critical relationship in the seq2seq model between hidden dimension size and working memory. For instance, does 128 approach average human capacity, or more generally might these numbers translate to the number of studied items held in working memory?

Thank you for this comment. In response, we have added the following clarifications to the Results section:

P14, L368: “Next, we use differences in hidden dimension size to model differences in working memory capacity. This view of working memory is consistent with previous modeling work in which recurrent neural activity is used to model active maintenance and integration of information^{2,51} and work in which capacity is affected by the number of hidden units in the recurrent neural networks⁵². Larger hidden dimension sizes allow more information to be held in the hidden state, though this only provides a relative measure of working memory – hidden dimension size does not directly translate to a specific number of items that can be held in working memory in human participants. We hypothesize that an increase in the model’s working memory capacity (as modeled by hidden dimension size) reduces the model’s need to obtain primacy through the attention mechanism.”

R2.6 - The authors describe a weakness of CMR as “it can determine the combination of the two contexts in a single parameter but cannot derive the dynamic interaction of these two contexts and their relation to free recall performance”.The authors explain what I think they refer to as the dynamic interaction by allowing dimension size to vary in the seq2seq model. If that is the case, in the CMR simulations the authors only explore one parameter set under one set of assumptions. I wonder if CMR were given the chance to vary in order to explain these dynamics as well, then it might account for dynamics like the seq2seq model.

Thank you for letting us know about this potential confusion. The experiments were to examine the role of hidden dimension size on the dynamic interaction of the two contexts. The dynamic interaction of the two contexts refers to how much context from the current step during each recall is coming from the previous time step and how much is coming from the reactivated/retrieved contexts from the encoding period ($c_j = \rho' c_{j-1} + \beta' [(1 - \gamma_{FC})x_j + \gamma_{FC} \alpha_j]$), which is governed by a single fixed parameter β' in CMR but by a dense layer to output different values (depending on circumstances of learning) in seq2seq models ($\hat{h}_j = \tanh(W_c [\phi(W_h' x_j + U_h' h_{j-1} + b_h'), \alpha_j])$). While it is possible to make CMR more dynamic by adding some method of varying this parameter in a dynamic way trained by reinforcement learning, doing so would essentially turn it into a neural network.

We have updated the text in the Discussions to clarify this:

P17, L488: “Recalled items during a free recall task, as specified by the CMR model, are driven by the current context, which is a combination of an evolving context from the previous time step (as maintained from the working memory) and a context reactivated from the encoding period (as retrieved from episodic memory, analogous to the attention mechanism). The CMR model can determine the combination of the two contexts in a single and fixed parameter, but cannot derive the dynamic interaction of these two contexts over time and their relation to free recall performance. Unlike CMR, the seq2seq

model has in place of this single fixed parameter a learned dense layer that can dynamically combine the two contexts in order to drive recall performance. This process helps disentangle the contributions of two types of contexts and identify their changes over each time step of the recall sequence. We showed that seq2seq models with smaller hidden dimensions, analogous to a limited working memory capacity, exhibit different recall strategies and a stronger reliance on the attention mechanism to access primacy items compared to models with larger hidden dimensions. ”

R2.7 - For any figures where values on the x and y axes overlap, perhaps the authors could offset the values on the x-axis to clarify this overlap. I wonder if this is why the “With Attention - Optimized” dark blue line appears minimally in Figures 6E,F.

Thank you for the suggestion, Fig. 6D-F have been changed and are now more legible.

Figure 6. Comparing optimized model behavior under ablation of the attention mechanism with recall behavior of medial temporal lobe (MTL) amnesia patients. (A–C) Behavioral patterns of MTL amnesia patients demonstrating memory deficits on a free recall task, reproduced from Palombo, et al³⁶. Despite the small list size, MTL amnesia patients exhibit diminished recall performance and reduced ability to perform backward recall transitions (conditional response probability at -1 lag). (D–F) Behavioral patterns of the seq2seq model with and without attention (hidden dimension size 64), optimized or during intermediate training. The model without attention exhibits diminished recall performance and reduced ability to perform backward recall transitions (no backward contiguity at any stage of training as seen from the conditional response probability at -1 lag). (G–H) Behavioral patterns of models of varying hidden dimension sizes with and without attention. Models without attention can achieve the optimal recall behavior but require sufficiently large hidden dimensions.

R2.8 - The authors describe CMR as having a “latent contextual representation”, but isn’t the contextual representation explicitly defined (e.g., Equation 3)?

Thank you for the suggestion. We used the term ‘latent’ to simply represent an internal, unobserved mental state that must be inferred to explain observable behavior. We acknowledge that this could be a source of confusion and have removed this description without affecting the message of the paragraph:

P4, L99: “Many behavioral findings in free recall literature can be captured by a model where a **contextual representation** slowly drifts over time and is associated with to-be-remembered experiences^{17,19,20}. The context at the end of the encoding period is carried over to the recall period if the delay or the distractor task between study and recall is minimal. During recall, the **contextual representation** serves as a cue to drive a sequence of recalls. Different than the development of machine translation models, which has been driven by the need to perform well on the translation task, the development of context-based models in human memory has been driven by the need to account for behavioral patterns in human data and can be traced back to the Bower’s (1972) temporal context model^{45,46}.”

R2.9 - I got a bit lost as to why it is informative to characterize the development of optimal weights across epochs for the seq2seq model, as opposed to only the final results. Don’t the earlier epochs partially rely simply on how the model is initialized?

Thank you for raising this important issue. While it’s true that early epochs are influenced by model initialization, our goal in analyzing intermediate training stages is to visualize how optimal recall strategies emerge over time. In Figures 4D–F, we show that earlier epochs of the seq2seq model exhibit behavioral patterns more aligned with average human recall (such as recency and backward contiguity) before transitioning toward the optimized, forward-asymmetric strategy characteristic of the model’s final performance. Furthermore, although random weight initialization does affect recall, earlier checkpoints consistently display instances of recency and backward contiguity, suggesting that these are features of less optimized models and not merely random occurrences. We have added these clarifications to the Results section next to Figure 4:

P11, L336: “To assess sub-optimal model behavior, we evaluate the recall behavior of several intermediate checkpoints taken of the model throughout the optimization process (Figures 4D–F). Characterizing these intermediate training stages provides insight into how recall strategies emerge and change as the model learns. We show that while the fully optimized seq2seq model with attention demonstrates the same behavior as the optimal policy of CMR, exhibiting both primacy and forward contiguity (Figures 4A–C), intermediate training evaluations of the model (lighter green lines) exhibit recency (higher recall probability at the end of list in Figure 4D in early Epochs 0–2) and backward contiguity (positive transition probabilities at negative lags in Figure 4F) similar to those observed in averaged human recall behavior. Recency and backward contiguity quickly disappear as the model is optimized to further rely on forward recalls initiated from the beginning of the sequence. This trend is made more evident in Figure 4G–I, which displays the change in the model’s tendency to initiate recall from the end (the last three items of the list; Figure 4G), to initiate recall from the beginning of the list (the first three items of the list; Figure 4H), and to recall items in the backward direction (conditional response probability with -1 lag; Figure 4I). Although model behavior during intermediate training evaluations is partially influenced by model initialization, we find that patterns such as recency and backward contiguity reliably appear across training runs and initializations, suggesting that these are stable features rather than idiosyncratic fluctuations.”

R2.10 - On page 10 when the authors refer to the “dashed line in Figure 4A-C”, I think they mean 4D-F.

Thank you for spotting this error. We have fixed this reference.

— Reviewer 3 —

R3.1 - My main issue with clarity is due to the fact that the seq2seq and CMR use different notation, making it hard at times to relate similar mechanisms. I appreciate that figures 1 and 2 try to use similar colors to group related processes, but I think this could still be further improved. For instance, by making the meanings of the colors more explicit via figure legends, or to group sections of the model in Figure 2 into higher-level “modules”, which could be enclosed in a box. This would allow for a shared schematic between the two approaches at the level of modules in order to provide a higher-level perspective.

But perhaps most importantly, it was hard to follow Eqs 1-9 due to jumping back and forth between the different notations used in each model. Some potential solutions could be to a) walk through each approach separately before integrating them together or b) develop a unified notation for both, which can be translated into the typical domain-specific notation in the SI. Either way, the current structure requires keeping in mind both notations simultaneously, which hurt

my ability to appreciate some of the more subtle aspects of this theory integration and may reduce the accessibility of this work for non-specialized audiences.

Thank you for these suggestions. We agree that the notational differences between the seq2seq and CMR models created an unnecessary barrier to understanding, particularly when readers were asked to track corresponding components across both frameworks. In response, we restructured the relevant section so that each model's encoding and recall mechanisms were introduced separately. Once both models were fully described, we presented a unified comparison through a direct mapping of the context drift and context reinstatement mechanisms.

We also revised Figure 2 to improve visual clarity and emphasize functional parallels. We hope these changes make the descriptions of each of the independent models as well as the architectural parallels between the two models clearer and easier to follow.

P4, L132

Deriving a Detailed Mathematical Mapping

In addition to highlighting the parallels in the historical developments of seq2seq models and CMR, in this section, we provide a detailed mathematical mapping between the architectural components of the two models. As illustrated in Figure 2, we align the encoding and decoding (or recall) processes across both frameworks, detailing each step. Specifically, Figures 2A and 2C show the encoding phases for the seq2seq model and the CMR model, respectively, while Figures 2B and 2D depict their corresponding decoding and recall phases. We will first describe each model in detail, then derive the mathematical mapping between them. Our model descriptions closely follow and are equivalent to prior work of seq2seq models with attention⁸ and CMR^{17,19,31}, though the exact notation has been slightly modified to facilitate easy visualization and alignment of the two models.

Figure 2. Detailed mapping between components in the seq2seq model with attention and those in the CMR Model. (A) Seq2Seq Encoding Phase: At each timestep i , the encoder RNN integrates the input (f_i) with the previous hidden state (h_{i-1}) to produce a new hidden state (h_i). This new hidden state is then concatenated with all preceding hidden states in the matrix H . (B) Seq2Seq Decoding Phase: The decoding phase is initiated using the final encoder hidden state as the decoder’s previous hidden state. On each decoding step j , the decoder RNN takes the previous output (f_{j-1}) and hidden state (h_{j-1}) to produce a new hidden state (h_j). Attention weights, w_j^i , are first computed by comparing the current decoder state with all encoder hidden states (by matrix multiplication), then used to generate the attention context vector (α_j) via a weighted summation. The context vector and decoder hidden state are then combined to form the final decoding hidden state \hat{h}_j , which is finally used to generate the next recalled item via the retrieval rule. (C) CMR Encoding Phase: The CMR model updates context (c_i) at each timestep by integrating the previous context (c_{i-1}) and the current item features (f_i) via context drift. The experimental item-to-context association matrix M_{exp}^{FC} is then updated by applying the Hebbian learning rule with the presented item and the previous context. (D) CMR Recall Phase: Recall is initiated from the final encoding context from the encoding phase. On each retrieval step, the current context c_j is updated to incorporate the influence from the previous context c_{j-1} , as well as the previous item embedding and experimental encoding contexts, via context drift similar to the encoding stage. When an item is recalled, the corresponding encoding context α_j is reinstated by application of the experimental association matrix M_{exp}^{FC} to the just recalled item f_{j-1} . Color coding and sub-figure alignment highlight the functional correspondence between hidden state/context, input, attention/context reinstatement, and sequential updating in both models.

Seq2seq model with attention

Encoding Phase The encoding phase of the seq2seq model, pictured in Figure 2A, is analogous to the study phase in a memory experiment, where participants encode a list of items into an evolving mental context. Each input item f_i presented at timestep i is embedded into a dense vector $x_i \in \mathbb{R}^d$ using pre-trained embedding vectors (in our case, GloVe embeddings⁴⁸). The encoder RNN processes the sequence one item at a time, updating its hidden state $h_i \in \mathbb{R}^d$ at each time step according to the equation below:

$$h_i = \phi(W_h x_i + U_h h_{i-1} + b_h) \quad (5)$$

In Eqn. 5, W_h and U_h are learned parameter matrices, b_h is a learned bias vector, ϕ is a non-linear activation function such as

tanh, sigmoid, etc., and h_{i-1} is the hidden state from the previous timestep. In this process, the weight matrix U_h controls the degree to which the previous hidden state h_{i-1} is maintained in the current hidden state h_i , while the weight matrix W_h controls the degree to which the embedding of the newly presented item, x_i , is incorporated into the current hidden state.

In addition to updating the hidden state h_i , the model also stores the concatenation of all hidden states, $H = [h_1, h_2, h_3, \dots, h_L]$ where L is the number of steps/items during encoding. These states H later inform the decoding process to access the encoder states through the attention mechanism.

Decoding Phase The decoding phase, pictured in Figure 2B, is initialized with the final hidden state from the encoding phase, i.e., h_L . During each decoding step j (which we distinguish from encoding step i), the decoder RNN receives the embedding $x_j \in \mathbb{R}^d$ of the just-recalled item from the previous step $j-1$, along with the previous hidden state $h_{j-1} \in \mathbb{R}^d$:

$$h_j = \phi(W_{h'}x_j + U_{h'}h_{j-1} + b_{h'}) \quad (6)$$

Eqn. 6 is identical to Eqn. 5 used in the encoding phase but employs different parameters $W_{h'}$, $U_{h'}$, and $b_{h'}$ learned for the decoding phase.

During decoding, the attention mechanism (shown in the green box in Figure 2B) gives the decoder access to all encoder hidden states H via attention weights. These attention weights are obtained by calculating a score function between the current decoder hidden state h_j and each hidden state h_i from the encoding stage. We use the dot product as the score function as specified in Luong attention⁸. to avoid introducing additional parameters. After calculating scores for each encoder hidden state, the softmax operation is applied to obtain the attention weights w as shown below:

$$w_j^i = \frac{\exp(h_j^\top h_i)}{\sum_{i=1}^L \exp(h_j^\top h_i)} \quad (7)$$

where $h_j^\top h_i$ is the dot product, i.e., the similarity between the current decoder hidden state j and an encoder hidden state h_i . The attention weight w_j^i refers to how much attention the model should pay to each encoder hidden state h_i relative to all other encoding states at the decoding step j , which represents the importance of the encoder hidden state h_i to producing the output at the decoding step j .

Once the attention weights have been computed, the overall attention context vector (α_j) is obtained via a weighted sum of encoder states:

$$\alpha_j = \sum_{i=1}^L w_j^i h_i \in \mathbb{R}^d \quad (8)$$

Before an output is generated, a final hidden state vector \hat{h}_j is formed by combining the decoder hidden state and the attention context through a dense, mixing layer:

$$\hat{h}_j = \tanh(W_c [h_j; \alpha_j]) \quad (9)$$

Here, $W_c \in \mathbb{R}^{d \times 2d}$ are the learned parameters of the mixing layer, while $[h_j; \alpha_j]$ is the concatenation of the current decoder hidden state h_j and the attention context vector α_j .

An output, f_j , is generated at the decoding step j based on an output/retrieval rule ψ in conjunction with the final hidden state \hat{h}_j :

$$f_j = \psi(\hat{h}_j) \in \mathbb{R}^N \quad (10)$$

where $f_j \in \mathbb{R}^N$ is a one-hot column vector that is all zeros except at the position representing the item's identity, and N is the total number of possible items in the experiment.

Context Maintenance and Retrieval (CMR) model

Encoding Phase The encoding stage in a free recall task is the period during which participants study a list of items in preparation for later recall. During the encoding phase in the free recall task (pictured in Figure 2C), participants study a list of L items one after another (drawn from a total number of N possible items in the experimental word pool). The CMR model proposes that their context slowly drifts towards the memory representations of recently encountered experiences. The state of the context at time step $c_i \in \mathbb{R}^N$ is given by:

$$c_i = \rho c_{i-1} + \beta x_i \quad (11)$$

where $x_i \in \mathbb{R}^N$ is the retrieved context (or input embeddings) of the just-encoded item, $\beta \in [0, 1]$ is a parameter determining the rate at which context drifts toward the new context, and ρ is a scalar ensuring $\|c_i\| = 1$. The retrieved context x_i is further expressed as:

$$x_i = M_{pre}^{FC} f_i \quad (12)$$

where $M_{pre}^{FC} \in \mathbb{R}^{N \times N}$ represents item-to-context associations that existed prior to the experiment (initialized as an identity matrix, under the simplifying assumption that an item is only associated with its own context; see ¹⁹), and $f_i \in \mathbb{R}^N$ is a one-hot column vector that is all zeros except at the position that represents an item's identity. Therefore, $M_{pre}^{FC} f_i$ is the context previously associated with the presented item at encoding step i , which is simply f_i . Together, Eqn. 11 captures how CMR embeds each item into a gradually drifting context space, binding together items encountered close together in time and allowing the model to capture the temporal contiguity effects observed in free recall ³⁴.

In addition to updating the context c_i , CMR forms associations between items and the evolving context throughout the encoding phase to capture new learning in the experiment – through experimental item-to-context and context-to-item associations held in $M_{exp}^{FC} \in \mathbb{R}^{N \times N}$ and $M_{exp}^{CF} \in \mathbb{R}^{N \times N}$. These matrices will be useful later in the recall phase to reactivate the corresponding encoding context of a given item (M_{exp}^{FC}) or to retrieve an item corresponding to a given context (M_{exp}^{CF}). They are initialized to zero at the start of the experiment and are updated via the Hebbian outer-product learning rule. Specifically, when an item is encoded at timestep i , an association is formed between the previous context state c_{i-1} and the presented item f_i :

$$\Delta M_{exp}^{FC} = c_{i-1} f_i^\top \quad (13)$$

Similarly, the association from context to item, M_{exp}^{CF} , is updated according to:

$$\Delta M_{exp}^{CF} = f_i c_{i-1}^\top \quad (14)$$

Following equations 13 and 14, after all L items in a list have been studied, the final experimental association matrices before the start of the recall phase can be written as:

$$M_{exp}^{FC} = \sum_{i=1}^L c_{i-1} f_i^\top \quad (15)$$

$$M_{exp}^{CF} = \sum_{i=1}^L f_i c_{i-1}^\top \quad (16)$$

To illustrate M_{exp}^{FC} and M_{exp}^{CF} defined in Eqns. 15 and 16 with an example, consider a short list of $L = 3$ items during encoding $f_1^\top = [0, 1, 0, \dots]$, $f_2^\top = [1, 0, 0, \dots]$ and $f_3^\top = [0, 0, 1, \dots]$, which are associated through Hebbian learning with contexts at the preceding step c_0 , c_1 and c_2 respectively (c_0 denotes the context vector prior to any encoding). After all L items in the list are encoded, the resulting association matrices take the following form $M_{exp}^{FC} = [c_1, c_0, c_2, \dots]$, where each context vector is stored in the column corresponding to the encoded item's identity. Similarly, the context-to-item association matrix is given by $M_{exp}^{CF} = [c_1^\top; c_0^\top; c_2^\top; \dots]$, where each context vector is stored in the row corresponding to the encoded item's identity.

Recall Phase The recall phase of CMR, pictured in Figure 2D, begins with the final context vector carried over from the encoding phase. On each recall step j , the context vector c_j is updated to reflect both the influence from the just-recalled item

f_{j-1} (controlled by β') and the context from the previous timestep c_{j-1} (controlled by ρ'), similarly to how context drifts during the encoding phase in Eqn. 11:

$$c_j = \rho' c_{j-1} + \beta' [(1 - \gamma_{FC})x_j + \gamma_{FC}\alpha_j] \quad (17)$$

However, different than the encoding phase, the reactivated context from the just-recalled item f_{j-1} comprises not only its pre-experimental context $x_j = M_{\text{pre}}^{FC} f_{j-1}$ (i.e., input embeddings, black arrow in Figure 2D), similarly to Eqn. 12, but also the experimental context α_j (green arrow in Figure 2D) which is given by:

$$\alpha_j = M_{\text{exp}}^{FC} f_{j-1} \quad (18)$$

The experimental context α_j retrieves the context associated with the item f_{j-1} during encoding through applying the experimental item-to-context associations M_{exp}^{FC} . The extent of retrieving an item's pre-experimental context x_j versus experimental context α_j , as shown in Eqn. 17, is determined by a parameter, $\gamma_{FC} \in [0, 1]$.

To determine which item f_j is to be retrieved at timestep j , CMR examines how much the current context c_j matches with all items' experimental contexts, as stored in rows of $M_{\text{exp}}^{CF} = \sum_{i=1}^L f_i c_{i-1}^\top$ (Eqn. 16):

$$a_j = M_{\text{exp}}^{CF} c_j = \left(\sum_{i=1}^L f_i c_{i-1}^\top \right) c_j = \sum_{i=1}^L f_i (c_{i-1}^\top c_j) \quad (19)$$

Here, $a_j \in \mathbb{R}^N$ is the activation strength, where each element reflects the similarity between the context at which item i was encoded, c_{i-1} , and the present context c_j during recall. Intuitively, items whose original encoding context more closely matches the current context are more likely to be recalled. Items that did not appear during the encoding phase have zero activation strengths.

To translate the activation strength a_j into recall probabilities, the model applies a softmax function over the subset of elements in a_j corresponding to items that appeared during the encoding phase. The probability of recalling the item from the encoding step i , f_i , at recall step j is given by:

$$P(f_j = f_i) = \frac{\exp [k c_{i-1}^\top c_j]}{\sum_{i'=1}^L \exp [k c_{i'-1}^\top c_j]} \quad (20)$$

Here, k is a parameter that determines the amount of noise present in the retrieval process, where a higher value of k favors a more noiseless recall and a higher chance of retrieving the item with the strongest activation. Finally, the recalled item at timestep j , f_j , is sampled from a multinoulli distribution whose probabilities for possible outcomes are specified in Eqn. 20.

Putting everything together: Mapping the seq2seq model with attention to CMR

Encoding Phase Both the encoding and decoding/recall phases of the seq2seq model with attention and CMR exhibit deep functional parallels. The correspondence between the two models during the encoding phase can be easily seen from Eqn. 5 and Eqn. 11. During each step of the encoding phase, both models update their current hidden state h_i or context state c_i by taking in an input embedding, x_i , that incorporates pre-experimental information of the presented item f_i , which is then combined with the hidden state h_{i-1} or context state c_{i-1} from the previous encoding step. In CMR, two parameters, ρ and β , control the mixing of the previous context and the new item embedding, while the seq2seq model accomplishes the same task through more complex parameter matrices W_h and U_h together with the bias vector b_h and the activation function ϕ .

Decoding/recall Phase Compared with the encoding phase, the parallels between the seq2seq model and CMR for the decoding phase are less straightforward and require additional derivations, which we will demonstrate in detail below. The key function during the decoding phase for both models is to generate an output or recall at each timestep j . While the seq2seq model with attention uses the hidden state \hat{h}_j to determine the output item f_j (as shown in Eqn. 10 and Figure 2B), the CMR model utilizes its context state c_j to determine the recalled item f_j (as shown in Eqn. 20 and Figure 2D). Our primary goal here is to demonstrate the correspondence in the decoding phase between these two models, specifically by showing that \hat{h}_j in the seq2seq model contains components equivalent to those of c_j in CMR.

Combining Eqn. 5 and Eqn. 9, \hat{h}_j in the seq2seq model with attention can be written as:

$$\hat{h}_j = \tanh (W_c [\phi (W_{h'} x_j + U_{h'} h_{j-1} + b_{h'}) ; \alpha_j]) \quad (21)$$

According to Eqn. 17, c_j in CMR is given by:

$$c_j = \rho' c_{j-1} + \beta' [(1 - \gamma_{FC})x_j + \gamma_{FC}\alpha_j] \quad (22)$$

Examining Eqn. 21 and Eqn. 22, it is clear that both \hat{h}_j and c_j share two common components: the hidden state or context state from the previous decoding step (h_{j-1} or c_{j-1}) and an input embedding (x_i) of the most recently recalled item. Crucially, c_j and \hat{h}_j also incorporate a third component, α_j , which are reactivated hidden states or contexts from the encoding phase. For the remainder of the derivation, we will show that these reactivated hidden states α_j^{RNN} (from the seq2seq model's attention mechanism) are equivalent to the reactivated contexts α_j^{CMR} (from the CMR model's context reinstatement mechanism).

Combining Eqn. 15 and Eqn. 18, we can write the reactivated context in CMR as:

$$\alpha_j^{CMR} = M_{\text{exp}}^{FC} f_{j-1} = \left(\sum_{i=1}^L c_{i-1} f_i^\top \right) f_{j-1} \quad (23)$$

Under the condition that the most recently recalled item at the decoding step $j - 1$ is the item studied at the encoding step i , i.e., $f_{j-1} = f_i$, Eqn. 23 can be written as:

$$\alpha_j^{CMR} = \left(\sum_{i=1}^L c_{i-1} f_i^\top \right) f_i = c_{i-1} \quad \text{if } f_{j-1} = f_i \quad (24)$$

Since item recall is probabilistic, the identity of the just-recalled item f_{j-1} is drawn from the distribution defined by the previous context state c_{j-1} , which can be written as (following Eqn. 20):

$$P(f_{j-1} = f_i) = \frac{\exp [k c_{i-1}^\top c_{j-1}]}{\sum_{i'=1}^L \exp [k c_{i'-1}^\top c_{j-1}]} \quad (25)$$

We can write the expected value of α_j^{CMR} as below, combining Eqn. 24 and Eqn. 25:

$$\mathbb{E}[\alpha_j^{CMR}] = \sum_{i=1}^L \frac{\exp (k c_{i-1}^\top c_{j-1})}{\sum_{i'=1}^L \exp (k c_{i'-1}^\top c_{j-1})} c_{i-1} \quad (26)$$

The expected value of context reinstatement in CMR, $\mathbb{E}[\alpha_j^{CMR}]$, is directly analogous to the attention context vector in the seq2seq model. Combining Eqn. 7 and Eqn. 8, we can write α_j^{RNN} in the seq2seq model as:

$$\alpha_j^{RNN} = \sum_{i=1}^L w_j^i h_i = \sum_{i=1}^L \frac{\exp (h_j^\top h_i)}{\sum_{i'=1}^L \exp (h_j^\top h_{i'})} h_i = \sum_{i=1}^L \frac{\exp (h_i^\top h_j)}{\sum_{i'=1}^L \exp (h_{i'}^\top h_j)} h_i \quad (27)$$

Examining Eqn. 26 against Eqn. 27, we can establish that the seq2seq model's attention mechanism is equivalent to the CMR model's context reinstatement mechanism, i.e., $\mathbb{E}[\alpha_j^{CMR}] \approx \alpha_j^{RNN}$. Intuitively, both models use a current decoding state (h_j or c_{j-1}) as a probe to reactivate a weighted average of states from the encoding phase. This retrieved context allows the model to access relevant information in the past to guide the generation of the next item. This concludes our proof for the decoding phase, where we demonstrate that what drives the next recall or output, \hat{h}_j in the seq2seq model and c_j in CMR, contains equivalent components as specified in Eqn. 21 and Eqn. 22."

R3.2 - One of the main findings is that seq2seq outperforms CMR in fitting human data. However, in which data regimes would this hold true? For instance, if you gave seq2seq less training data, would the performance cross-over at some point below CMR? Thus, it would be important to more carefully quantify the interpretation of the results to run analyses using different amounts of training data to see where the boundary conditions are.

Thank you for raising this important point. We agree that it was essential to assess the conditions under which the seq2seq model outperforms CMR, particularly in data-limited settings. To investigate this, we conducted additional analyses by retraining the individual subject models using reduced training data: specifically, 60-20-20 and 40-30-30 train-validation-test splits (compared with 90-5-5 originally). With our available hardware, training a seq2seq model to fit a single subject takes on average 3.5 hours, while fitting CMR to a single subject takes approximately 4 hours. Due to the size of the dataset (171 subjects), we randomly sampled 20 subjects from the full cohort of 171 for these evaluations. We added the full analyses in a new section in the Supplementary Materials “Individual Subject Fitting Under Different Training Dataset Sizes”:

SP2, L34: “To assess how the amount of training data influences the ability of the seq2seq model (with attention) to outperform CMR in fitting individual-level recall data, we compared the fit quality of both models across different training data sizes. We trained and evaluated models on 20 randomly selected subjects using three different train-validation-test splits: 90-5-5, 60-20-20, and 40-30-30. For each split, models were fit to individual subjects and assessed on root-mean-square error (RMSE) across the three sets of recall behavior metrics (the serial position curve, the probability of first recall, and the conditional response probability). Figure S3A displays the average RMSE across all tasks for each data split across all 20 subjects included in the study.

Results show that whether the seq2seq model outperforms CMR depends on the training data size. Pictured in Figure S3B-D are the distributions of RMSE values for the serial position curve, the probability of first recall, and the conditional response probability, respectively, across all splits. In the 90-5-5 regime, the seq2seq model significantly outperformed CMR, yielding lower RMSE across subjects (Wilcoxon signed-rank test: two-sided: $W = 8.0, n = 20, p = 4.77 \times 10^{-5}$). However, as the amount of training data decreased, this advantage was reversed. In both the 60-20-20 and 40-30-30 splits, CMR significantly outperformed the seq2seq model (60-20-20: two-sided: $W = 0.0, n = 20, p = 1.91 \times 10^{-6}$; $W = 210.0, n = 20, p = 9.54 \times 10^{-7}$; 40-30-30: two-sided: $W = 5.0, n = 20, p = 1.91 \times 10^{-5}$). This crossover indicates that CMR achieves better fits than seq2seq when the available training data is limited. These results support the conclusion that neural network models, such as our seq2seq model, can surpass traditional models given ample training data, but established cognitive models like CMR are more efficient in scenarios where training data is limited. ”

We also added a summary of these analyses in the main Results section:

P10, L 299: “When the amount of training data is reduced, however, CMR is shown to give better prediction, highlighting a trade-off between model flexibility and data efficiency (see additional model comparisons in Supplementary Materials S3).”

Figure S3: Individual subject fitting results for the seq2seq model with attention and CMR across different amounts of training data. (A) Each grouped set of bars represents the root-mean-square error (RMSE) between the seq2seq model with attention or the CMR model and the human subject data for a given data split for each subject. The seq2seq model with attention exhibits lower RMSE values than CMR with ample training data (90-5-5 split) but higher RMSE values with less training data (60-20-20 and 40-30-30 splits). (B-D) Distributions of RMSE values for the serial position curve, the probability of first recall, and the conditional response probability, respectively, across all splits.

R3.3 - Currently, the results make this difficult to assess. Details are missing about how cross-validation is performed. The statistics on page 8 lack DOFs. And why does Figure 3 present these specific 12 subjects?

Thank you for the suggestions. We have provided more details on how data is split in the Methods section:

P18, L571: “**Explain and Predict Individual Subject Behavior** - For individual subject fitting, each subject’s set of presentation–recall pairs was partitioned into training (90%), validation (5%), and test (5%) splits. Data splitting was performed randomly, ensuring that no presented list appeared in more than one subset. Seq2seq models with attention and a hidden dimension size of 128 were trained on the training set, with generalization loss evaluated on the validation set and final performance reported on the held-out test set, as presented in the main text. This individual fitting training was performed using a standard cross-entropy loss due to the order of recall being necessary to capture the recall characteristics of each subject. [...] For comparison, a CMR model was fit using Bayesian optimization⁸⁰ for a total of 300 optimization iterations on each subject’s training and validation splits.”

Although we did not use cross validation, we run additional cross-validation analyses to show that the same conclusion holds while using 5-fold cross-validation compared with the original 90-5-5 split. Performing this analysis on subjects was impractical given time and compute constraints. A full 5-fold cross-validation run for a single subject requires on average 17.5 hours of training time. Extending this procedure to all 171 subjects in our dataset would require over 3,000 GPU-hours (i.e. almost 125 days), making a full cross-validation analysis of the entire dataset infeasible with our resources and time limitations. To address this, we have randomly chosen 20 subjects from the original dataset on which to perform the 5-fold cross-validation study. The results added to the supplementary material are as follows:

SP1, L12: “**Cross-validation Study of Individual Subject Fitting** - To assess whether different methods of conducting the

training and testing splits would affect our conclusion that seq2seq performs better than CMR in predicting recall behavior, we conducted a 5-fold cross-validation on a random subset of 20 subjects from the full dataset of 171 subjects given our time and compute constraints (A full 5-fold cross-validation run for a single subject requires on average 17.5 GPU-hours). In this cross-validation, the dataset is randomly divided into five equal parts, and the model is trained on four parts while the remaining part is used for validation and testing; this process is repeated five times so that each part serves as the validation and test set once. To maintain consistency with our original 90%-5%-5% train-validation-test splits, we used the standard 5-fold cross-validation with an 80%-20% train-test split. After designating 80% of the data for training in each fold, we randomly sampled two non-overlapping 5% subsets – one for validation and one for testing – from the remaining 20% of data. The seq2seq model with attention was trained and evaluated using the same procedures as in the main analysis. Model performance was assessed on held-out folds using root-mean-square error (RMSE) across the three behavioral measures (the serial position curve, the probability of first recall, and the lag-conditional response probability) as in our original analysis depicted in Figure 3. Figure S2 shows the RMSE for each cross-validation fold relative to the corresponding subject’s recall patterns (averaged across all behavioral measures), with the RMSE for CMR included for comparison.

We see that a significant difference between seq2seq and CMR RMSE remains after the cross-validation study, corroborating the results of our main analysis. To ensure this difference was significant, we conducted a Wilcoxon signed-rank test comparing the average of the RMSE means (i.e., the average RMSE for each of the three behavior measures, averaged across all seq2seq model splits) with the CMR RMSEs (two-sided: $W = 0, n = 20, p = 1.907 \times 10^{-6}$). This result indicates that the seq2seq models exhibit significantly lower RMSE values on average compared to CMR. These findings support that different methods of conducting the training and testing splits do not affect our conclusions in model comparison.

A mention in the main text has been added as follows:

P10, L298: “These results are not affected by different methods of conducting the training and testing splits (see Supplementary Materials S2).”

Figure S2: 5-Fold cross-Validation results of individual subject fitting for the seq2seq model with attention and CMR. Model performance was assessed on held-out folds using root-mean-square error (RMSE) across all three behavioral measures (the serial position curve, the probability of first recall, and the lag-conditional response probability).

For the Mann-Whitney tests on page 8, we have reported n_1 and n_2 but not df . This is because degree of freedom does not apply to non-parametric tests. However, during our revision, we have shifted all Mann-Whitney tests to the Wilcoxon signed-rank test (another non-parametric test but for paired observations), which is more appropriate for pairing the seq2seq and CMR data per subject; for these tests, we now report the number of paired samples (n) rather than n_1 and n_2 . None of the conclusions have changed after shifting to the Wilcoxon signed-rank test.

We have changed the tests in the following locations:

P10, L298: We find that the seq2seq model fits the corresponding human data curve with significantly smaller error for serial position (Wilcoxon signed-rank test: two-sided, $W = 0.0$, $n = 171$, $p = 8.2 \times 10^{-30}$), probability of first recall (Wilcoxon signed-rank test: two-sided, $W = 2880.0$, $n = 171$, $p = 5.23 \times 10^{-12}$), and conditional response probability (Wilcoxon signed-rank test: two-sided, $W = 3822.0$, $n = 171$, $p = 5.51 \times 10^{-8}$). These results are not affected by different methods of conducting the training and testing splits (see Supplementary Materials S2).

SP1, L28: To ensure this difference was significant, we conducted a Wilcoxon signed-rank test comparing the average of the RMSE means (i.e., the average RMSE for each of the three behavior measures, averaged across all seq2seq model splits) with the CMR RMSEs (two-sided: $W = 0$, $n = 20$, $p = 1.907 \times 10^{-6}$). This result indicates that the seq2seq models exhibit significantly lower RMSE values on average compared to CMR. These findings support that different methods of conducting the training and testing splits do not affect our conclusions in model comparison.

SP2, L46: In the 90-5-5 regime, the seq2seq model significantly outperformed CMR, yielding lower RMSE across subjects (Wilcoxon signed-rank test: two-sided: $W = 8.0$, $n = 20$, $p = 4.77 \times 10^{-5}$). However, as the amount of training data decreased, this advantage was reversed. In both the 60-20-20 and 40-30-30 splits, CMR significantly outperformed the seq2seq model (60-20-20: two-sided: $W = 0.0$, $n = 20$, $p = 1.91 \times 10^{-6}$; 40-30-30: two-sided: $W = 5.0$, $n = 20$, $p = 1.91 \times 10^{-5}$). This crossover indicates that CMR achieves better fits than seq2seq when the available training data [...]

Finally, with respect to Figure 3, the 12 subjects shown were the first 12 participants (by index) from the PEERS dataset. These were not hand-selected based on any aspect of their behavior or model fit. To avoid any impression of selection bias, we have added clarifications both in main results “Figures 3A–C depict the behavior patterns of the first 12 subjects (for illustrative purposes) of the 171 subjects taken from the PEERS dataset with both fitted CMR and seq2seq models overlaid” as well as in the Figure 3 caption “Behavioral patterns for the first 12 subjects out of 171 subjects”.

R3.4 - Additionally, Figure 4D is never explained in the text, even though I found this to be one of the most interesting results. There may also be some other mistakes in figure references, since the text jumps from referencing Figure 4c to Figure 4G, omitting the entire 2nd row. On page 10 there is a reference to dashed lines in Figure 4A-C, but there are no lines.

Thank you for pointing it out. We missed explicit figure references although we discussed Figure 4D and Figure 4D-F (the second row) in the results. We have now added these explicit figures references to the Results section. Additionally, we’ve corrected the unrelated figure reference issue you mentioned regarding dashed lines in Figure 4A–C, which no longer appear in the final version.

P11, L332: “Only a small proportion of top-performing human participants can demonstrate the exact behavior of the optimal policy³¹. Averaged human recall behavior in free recall experiments (dashed line in Figure 4D–F; reproduced from Kahana et al.⁴⁹) also exhibits recency (enhanced end-of-list recall) and backward contiguity (tendency toward shorter than longer backward lags), in addition to what is amplified in the optimal behavior with primacy (enhanced beginning-of-list recall) and forward contiguity (tendency toward shorter than longer forward lags). To assess sub-optimal model behavior, we evaluate the recall behavior of several intermediate checkpoints taken of the model throughout the optimization process (Figures 4D–F). Characterizing these intermediate training stages provides insight into how recall strategies emerge and change as the model learns. We show that while the fully optimized seq2seq model with attention demonstrates the same behavior as the optimal policy of CMR, exhibiting both primacy and forward contiguity (Figures 4A–C), intermediate training evaluations of the model (lighter green lines) exhibit recency (higher recall probability at the end of list in Figure 4D in early Epochs 0–2) and backward contiguity (positive transition probabilities at negative lags in Figure 4F) similar to those observed in averaged human recall behavior. Recency and backward contiguity quickly disappear as the model is optimized to further rely on forward recalls initiated from the beginning of the sequence. This trend is made more evident in Figure 4G–I, which displays the change in the model’s tendency to initiate recall from the end (the last three items of the list; Figure 4G), to initiate recall from the beginning of the list (the first three items of the list; Figure 4H), and to recall items in the backward direction (conditional response probability with -1 lag; Figure 4I). Although model behavior during intermediate training evaluations is partially influenced by model initialization, we find that patterns such as recency and backward contiguity reliably appear across training runs and initializations, suggesting that these are stable features rather than idiosyncratic fluctuations.”

The authors did a thorough job addressing my suggestions and concerns. I understand that performing iterative cross-validation on the entire dataset would be computationally too demanding. My only minor remark concerns Figure 3F, where the forward asymmetry, a key feature in the human data, appears to be nearly gone in Seq2Seq.

Thank you for drawing attention to this point. We see from Figure 3C that the Seq2Seq is capable of capturing both forward and backward asymmetry, and we believe that this relative lack of asymmetry in the averaged plot (Figure 3F) is a scaling issue in the plot that makes it difficult to see the level of asymmetry across the averaged curves. We have updated the size and scaling of the subplots in Figure 3 to help make these trends more noticeable.

Figure 1. Behavioral patterns for individual participants and the model predictions. (A–C) Behavioral patterns for the first 12 participants out of 171 participants, reproduced from Experiment 1 of the PEERS free recall dataset², overlaid with behavioral patterns of the CMR model and the seq2seq model with attention, trained over a separate subset of the same individual participant data. The behavioral patterns include (A) the serial position curve, (B) the probability of the first recall, and (C) the conditional response probability. (D–F) Aggregated behavioral patterns for all 171 participants overlaid with aggregated CMR and seq2seq model behavioral patterns. Model fits are quantitatively evaluated regarding the root-mean-square error between the human behavioral patterns of the 171 participants and model predictions over (G) the serial position curve, (H) the probability of the first recall, and (I) the conditional response probability. Both Seq2Seq and CMR model fitting analyses are for $N = 171$ participants.

— Reviewer 2 —

I am grateful to the authors for addressing my concerns and clarifying my points of confusion. After reviewing these points as well as their other revisions, I recommend the manuscript for publication.

— Reviewer 3 —

My concerns relating to clarity have been greatly improved in the revised version of the manuscript, particularly with the new Fig. 2. The updated equations certainly help as well and are quite thorough, although the authors may decide on their own whether to move some elements to the methods or SI to prevent hurting the flow of the paper.

I'm also very happy with how the authors addressed my concern about the generalizability of the results in terms of the amount of training data. One very minor point about Figure S3 is that in panels B–D it's difficult to tell which model wins. Fig. S2 is also quite difficult to assess, and it might be better to plot some aggregate means (e.g., boxplots by split) rather than the current bar plots separated by subject.

All my other concerns have also been addressed in the revision.

Thank you for pointing out these concerns with the supplementary figures. For Figure S2, we have converted the figure into a set of box-and-whisker plots to allow the individual data splits to be compared more easily.

Figure 2. 5-fold cross-validation results of individual participant fitting for the seq2seq model with attention and CMR. Model performance was assessed on held-out folds using root-mean-square error (RMSE) across all three behavioral measures (the serial position curve, the probability of first recall, and the conditional response probability). ($N = 20$ for all analyses.)

Additionally, we have added explicit indicators (vertical dashed lines) for the mean of each RMSE distribution in Figures S3B–D to help readers better compare each of the models/data splits.

Figure 3. Individual participant fitting results for the seq2seq model with attention and CMR across different amounts of training data. (A) Each grouped set of bars represents the root-mean-square error (RMSE) between the seq2seq model with attention or the CMR model and the human participant data for a given data split for each participant. The seq2seq model with attention exhibits lower RMSE values than CMR with ample training data (90–5–5 split) but higher RMSE values with less training data (60–20–20 and 40–30–30 splits). (B–D) Distributions of RMSE values for the serial position curve, the probability of first recall, and the conditional response probability, respectively, across all splits.

— Reviewer 4 —

The authors of this manuscript compare the similarities in architecture of a seq2seq model with an attention mechanism and a CMR model. Their ultimate goal is to use this novel seq2seq model in human cognition, namely memory. They showcase the similarities in task performance between the two model types in a list learning task compared to human performance, quantify the relationship between attention/primacy effects and working memory capacity, and leverage the attention mechanism in the seq2seq model to possibly explain task performance for amnesia patients. Together, these results show that a seq2seq model with an attention mechanism is similar to a CMR model’s context mechanism, suggesting that this type of seq2seq model can be used to accurately model human memory in both healthy controls and those with MTL-based amnesias.

The way that the authors break down the similarities in the architectures of their proposed RNN and a CMR model was very easy to follow. The list-learning task clearly showed how their RNN model with attention performs similarly to a CMR model. One of the most interesting uses of the model was the simulation of memory deficits in amnesia, which shows that this model is useful not for just modeling healthy controls but also for patient groups. In reviewing the comments to the previous submission of this manuscript, the authors have been very thorough in their responses, which have made their work extremely clear. I have no additional comments, and this manuscript will be a great addition to the literature on the use of computational models to explain human memory.

The code is very well commented out with an easy to follow README document. It functions well and is very detailed, which helps to easily reproduce their results.